# Structural insights into regulation of CNNM-TRPM7 divalent cation uptake by the small GTPase ARL15

Luba Mahbub[1,2†], Guennadi Kozlov[1,2†], Pengyu Zong[3], Emma L Lee[1,2], Sandra Tetteh[4], Thushara Nethramangalath[4], Caroline Knorn[1,2], Jianning Jiang[1,2], Ashkan Shahsavan[1,2], Lixia Yue[3], Loren Runnels[4], Kalle Gehring[1,2]*

[1]Department of Biochemistry, McGill University, Montreal, Canada; [2]Centre de recherche en biologie structurale, McGill University, Montréal, Canada; [3]Department of Cell Biology, UCONN Health Center, Farmington, United States; [4]Rutgers-Robert Wood Johnson Medical School, Piscataway, United States

**Abstract** Cystathionine-β-synthase (CBS)-pair domain divalent metal cation transport mediators (CNNMs) are an evolutionarily conserved family of magnesium transporters. They promote efflux of $Mg^{2+}$ ions on their own and influx of divalent cations when expressed with the transient receptor potential ion channel subfamily M member 7 (TRPM7). Recently, ADP-ribosylation factor-like GTPase 15 (ARL15) has been identified as CNNM-binding partner and an inhibitor of divalent cation influx by TRPM7. Here, we characterize ARL15 as a GTP and CNNM-binding protein and demonstrate that ARL15 also inhibits CNNM2 $Mg^{2+}$ efflux. The crystal structure of a complex between ARL15 and CNNM2 CBS-pair domain reveals the molecular basis for binding and allowed the identification of mutations that specifically block binding. A binding deficient ARL15 mutant, R95A, failed to inhibit CNNM and TRPM7 transport of $Mg^{2+}$ and $Zn^{2+}$ ions. Structural analysis and binding experiments with phosphatase of regenerating liver 2 (PRL2 or PTP4A2) showed that ARL15 and PRLs compete for binding CNNM to coordinate regulation of ion transport by CNNM and TRPM7.

**\*For correspondence:**
kalle.gehring@mcgill.ca

[†]These authors contributed equally to this work

**Competing interest:** The authors declare that no competing interests exist.

## Editor's evaluation

In this important work, Mahbub and colleagues examine how the small GTPase ARL15 regulates ion flux through the proposed CNNM-TRPM7 complex. Using a complementary array of techniques, the authors gathered solid evidence for the binding of ARL15 to CNNM, resulting in a proposal on how this may affect the function of the CNNM-TRPM7 complex.

## Introduction

Magnesium ($Mg^{2+}$) is the most abundant intracellular divalent cation and essential for key cellular processes such as energy production and protein synthesis. In order to maintain and regulate magnesium levels, cells possess a number of $Mg^{2+}$ channels and transporters. Among these are transient receptor potential ion channel subfamily M member 7 (TRPM7) and cystathionine-β-synthase (CBS)-pair domain divalent metal cation transport mediators (CNNMs). TRPM7 is a ubiquitously expressed ion channel with a C-terminal kinase domain also involved in $Ca^{2+}$ homeostasis and $Zn^{2+}$ transport (*Monteilh-Zoller et al., 2003*; *Clark et al., 2006*; *Wei et al., 2009*; *Visser et al., 2013*; *Krapivinsky et al., 2014*; *Mittermeier et al., 2019*). CNNMs are a widely conserved family of integral membrane proteins and mutated in genetic diseases linked to $Mg^{2+}$ uptake or transport (*Wang et al., 2003*; *Funato and Miki, 2019*; *Giménez-Mascarell et al., 2019*; *Chen and Gehring, 2023*).

Recently, TRPM7 and CNNMs were found to function together to mediate divalent cation influx as a trimeric complex with ADP-ribosylation factor-like GTPase 15 (ARL15) (*Zolotarov et al., 2021*; *Kollewe et al., 2021*; *Bai et al., 2021*). Mg$^{2+}$ uptake experiments demonstrated that CNNMs employ the TRPM7 channel and that, in the absence of the channel, CNNM2 and CNNM4 can lower intracellular Mg$^{2+}$ levels (*Bai et al., 2021*). Experiments showed that ARL15 binds the C-terminal portion of CNNM2 and increases CNNM3/TRPM7 protein complex formation to reduce TRPM7 activity (*Hardy et al., 2023*; *Zolotarov et al., 2021*). Despite recent advances in structural studies of TRPM7 (*Nadezhdin et al., 2023*; *Duan et al., 2019*), the structure of the CNNM-TRPM7 complex remains unknown.

ARL15 is a member of the RAS superfamily of small GTPases and associated with several metabolic traits including disorders of lipid metabolism (*Richards et al., 2009*; *Rocha et al., 2017*; *Wu et al., 2021*). Genome-wide association studies have found associations with coronary heart disease, kidney disease, rheumatoid arthritis, and diabetes (*Gorski et al., 2017*; *Negi et al., 2013*; *DIAbetes Genetics Replication And Meta-analysis (DIAGRAM) Consortium et al., 2014*). Within the GTPase superfamily, ARL15 is most closely related to ADP-ribosylation factor (ARF) and ARF-like (ARL) GTPases. ARF GTPases are often involved in membrane-trafficking pathways, while the subfamily of ARL proteins have more diverse functions (*Burd et al., 2004*; *Sztul et al., 2019*).

Structurally, CNNMs consist of an N-terminal extracellular domain, a transmembrane domain, and two cytosolic domains: a CBS-pair domain (also termed a Bateman domain) and a cyclic nucleotide-binding homology (CNBH) domain (*de Baaij et al., 2012*). Two structures of prokaryotic orthologs have been determined, confirming that the CNNMs are ion transporters (*Chen et al., 2021*; *Huang et al., 2021*). The CBS-pair and CNBH domains from human CNNMs have been extensively studied with multiple structures determined including complexes with phosphatases of regenerating liver (PRLs) (*Chen et al., 2018*; *Gulerez et al., 2016*; *Zhang et al., 2017*; *Chen et al., 2020*; *Corral-Rodríguez et al., 2014*; *Giménez-Mascarell et al., 2017*). CNNM CBS-pair domains bind PRLs with low nanomolar affinity through an aspartic acid that inserts into the phosphatase catalytic site, mimicking a phosphoprotein substrate (*Gehring et al., 2022*). The binding regulates Mg$^{2+}$ transport and promotes tumor progression in cancer (*Funato et al., 2014*; *Hardy et al., 2015*).

Here, we used a variety of biophysical techniques to study the interaction of ARL15 with CNNM proteins. Isothermal titration calorimetry (ITC) and NMR experiments demonstrate that ARL15 binds GTP with micromolar affinity, orders of magnitude weaker than typical GTPases. We show that ARL15 binds CNNM CBS-pair domains with low micromolar affinity and inhibits CNNM2 Mg$^{2+}$ efflux. We present the crystal structure of the complex of ARL15 with the CNNM2 CBS-pair domain. The structure shows that the ARL15-binding site overlaps with the previously identified PRL-binding site. ITC and NMR experiments confirm that the proteins compete for CNNM binding. The structure also allowed us to design mutants that specifically disrupt ARL15-CNNM binding. Loss of CNNM binding leads to a loss in the ability of ARL15 to suppress CNNM2 and TRPM7 channel activity in cultured cells, confirming the biological relevance of the molecular interaction.

## Results

### ARL15 binds GTP with micromolar affinity

We prepared recombinant ARL15 protein using bacterial expression. Full-length ARL15 showed a propensity to aggregate that hampered its analysis so a truncated version encompassing only the ARL15 GTPase domain (residues 32–197) was prepared. The N-terminal residues removed are the site of ARL15 palmitoylation and promote membrane binding (*Wu et al., 2021*), while C-terminal residues removed have no known function. The truncated protein was stable and soluble at high micromolar concentrations.

Initial attempts to preload ARL15 with GTP or GDP were unsuccessful and suggested that ARL15 has weak binding affinity and was nucleotide-free when purified from *Escherichia coli*. To confirm this, we compared its UV spectrum with that of hRAS, a well-characterized high affinity nucleotide binding GTPase (*Figure 1A*). The UV spectrum of hRAS is flat absorbance from 250 to 290 nm due to the presence of bound nucleotide while ARL15 shows a deep dip around 250 nm. We heat denatured the samples to separate the proteins and bound nucleotides (*Figure 1—figure supplement 1A*). The ARL15 sample showed no absorbance after protein precipitation while the hRAS spectrum revealed the bound nucleotide.

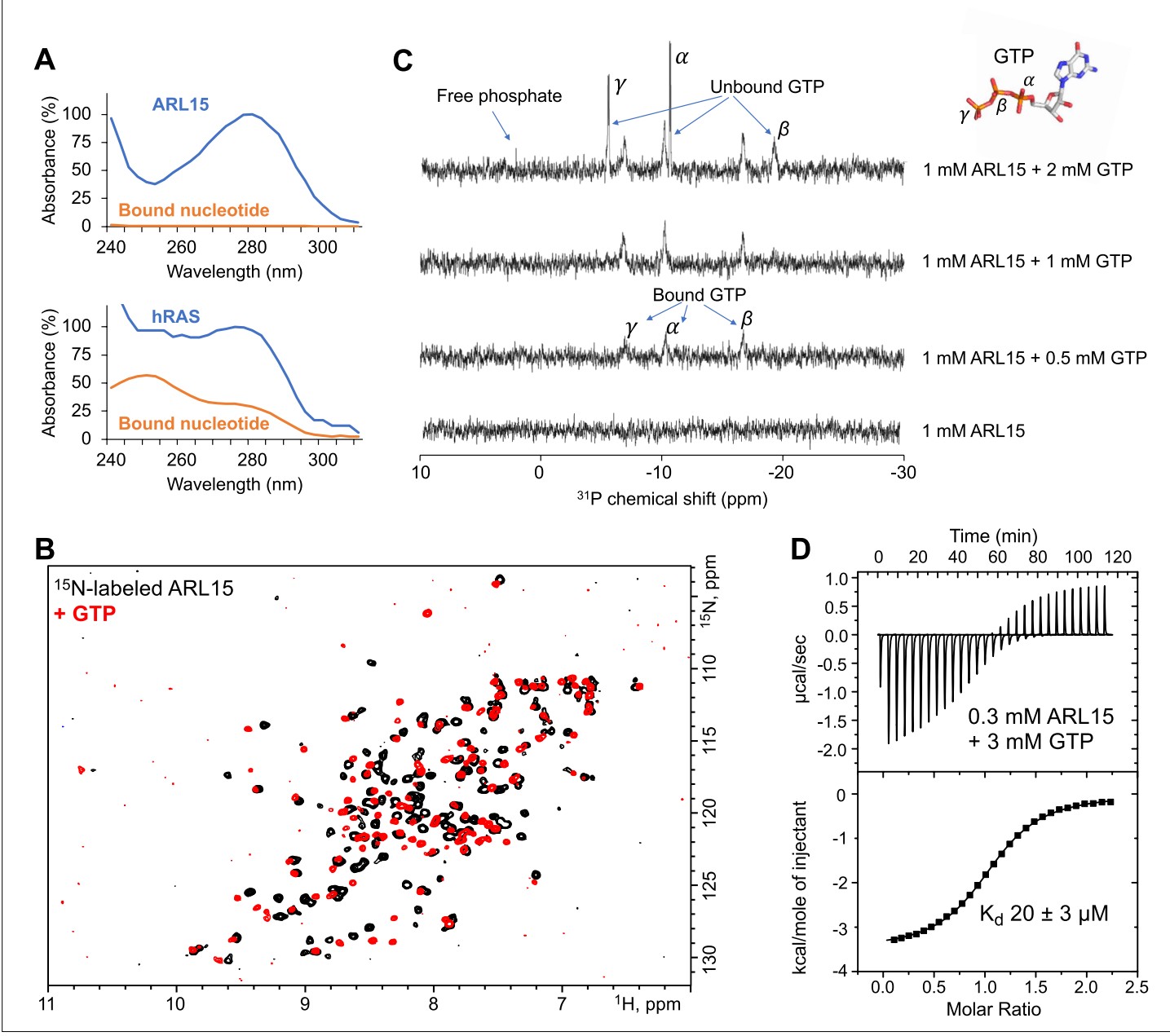

**Figure 1.** ADP-ribosylation factor-like GTPase 15 (ARL15) (32–197) purifies without nucleotide bound and has weak affinity for GTP. (**A**) UV spectra of recombinant ARL15 and the GTPase hRAS. Spectra were recorded before (*blue*) and after heat treatment (*orange*) to precipitate the protein component. Unlike hRAS, ARL15 has no nucleotide bound. (**B**) $^1$H-$^{15}$N correlation NMR spectra of 100 μM ARL15 unliganded (*black*) and with 500 μM GTP (*red*). (**C**) Phosphorus ($^{31}$P) NMR spectra of recombinant ARL15 titrated with GTP. Prior to addition of GTP, no signals are visible. GTP binds until ARL15 is saturated whereupon signals for both bound and free GTP are observed. (**D**) Isothermal titration calorimetry shows ARL15 binds GTP with mid-micromolar affinity.

The online version of this article includes the following source data and figure supplement(s) for figure 1:

**Source data 1.** Fitting parameters for isothermal titration calorimetry (ITC) thermogram of ADP-ribosylation factor-like GTPase 15 (ARL15) binding to GTP in *Figure 1D*.

**Figure supplement 1.** ADP-ribosylation factor-like GTPase 15 (ARL15) has low intrinsic GTPase activity.

We turned to NMR to characterize the GTP-binding properties of ARL15 and confirm the absence of bound nucleotide. 2D NMR of $^{15}$N-labeled ARL15 (32–197) showed well-dispersed signals of equal intensity, characteristic of a well-folded, homogenous protein preparation (*Figure 1B*). Titration with GTP caused changes in the spectrum in the slow-exchange NMR time scale and a general improvement in strength and uniformity of the signals.

To substantiate the absence of nucleotide in the purified recombinant ARL15, we acquired phosphorous NMR spectra before and after GTP addition (*Figure 1C*). No $^{31}$P NMR signals were observed in the initial spectrum. Upon addition of half an equivalent of GTP, three broad signals from bound GTP phosphorus atoms appeared. Upon over titration, additional sharper peaks appeared that correspond to free (unbound) GTP. To assess the intrinsic GTPase activity of ARL15, we incubated the sample overnight (*Figure 1—figure supplement 1B*). After 14 hr, a small signal from free phosphate was visible along with a 25% decrease in free GTP. Assuming the activity is not a contaminating bacterial enzyme, this corresponds to a turnover rate of $5 \times 10^{-6}$ s$^{-1}$ or roughly an order of magnitude smaller than the intrinsic GTPase activity of $7.1 \times 10^{-5}$ s$^{-1}$ of hRAS (*Neal et al., 1988*).

Next, we used ITC to measure the affinity of ARL15 for GTP. A clear and unambiguous thermogram was observed with 300 µM ARL15 that could fit a single binding site model with an affinity of 20 µM (*Figure 1D*). This is almost six orders of magnitude weaker than the picomolar affinity of hRAS and other small GTPases (*Ford et al., 2009*; *Neal et al., 1988*). The NMR and ITC experiments showed rapid exchange (binding) of nucleotide confirming that ARL15 is a highly atypical GTPase.

## ARL15 binds CNNMs with low micromolar affinity

To quantify ARL15 binding, we prepared constructs of CNNM2 containing the CBS-pair domain (residues 429–584) and a larger cytosolic fragment (429–817) containing the CNBH domain (*Figure 2A*). ITC experiments with both constructs generated a strong exothermic signal when titrated with ARL15 (32–197), corresponding to affinities of 1–2 µM (*Figure 2B*). Unlike previous co-immunoprecipitation experiments (*Zolotarov et al., 2021*), we did not observe a significant increase in affinity when the CNBH domain was present (*Figure 2C*). Experiments with the CBS-pair domains of CNNM3 and CNNM4 showed similar binding affinities; addition of the CNBH domains had no effect. ITC experiments with CBS-pair domain of CNNM1 also showed binding but with roughly 15-fold weaker affinity. These results demonstrate that ARL15 binding is a conserved property of CNNM proteins.

We next tested whether nucleotide-binding affects the interaction. Unexpectedly, CNNM binding was not affected by the absence or presence of GTP, while GDP addition had a minor effect, raising the $K_d$ by about 40% (*Figure 2D*). These results suggest that the ARL15 nucleotide-binding switch regions are not significantly involved in CNNM binding. We also tested if Mg$^{2+}$ or ATP affects ARL15 binding. The addition of 5 mM EDTA to the binding buffer (which contained 1 mM Mg$^{2+}$) had no effect on binding in agreement with co-immunoprecipitation experiments (*Zolotarov et al., 2021*). On the other hand, we observed a small, twofold improvement in affinity when ATP was present. To confirm that the effect was due to ATP binding to the CBS-pair domain, we repeated the binding measurements with a pathogenic CNNM2 mutation (T568I) that blocks ATP binding (*Hirata et al., 2014*). The mutant bound ARL15 as well as the wild-type (WT) domain in the absence of ATP, but it did not show increased affinity in the presence of ATP. This confirmed that a small amount of positive allosteric coupling exists between the CNNM2 ATP and ARL15-binding sites.

## Structure of the ARL15-CNNM CBS-pair domain complex

We turned to X-ray crystallography to visualize the ARL15-CNNM interaction (*Figure 3* and *Figure 3—video 1*). Commercial crystallization screens were used for mixtures of ARL15 (32–197) with cytosolic or CBS constructs of CNNM2, CNNM3, and CNNM4. Small crystals were obtained with a sample containing ARL15 (32–197) and CNNM2 CBS-pair domain (429–584) (*Figure 3A–B*). The crystals were improved by moving the His-tag on ARL15 to the C-terminus and inclusion of a non-hydrolyzable GTP analog, GppNHp. The best crystals diffracted to 3.2 Å using synchrotron radiation (*Table 1*). The structure was solved by molecular replacement using the structure of the CNNM2 CBS-pair domain (PDB 4IY0) and an AlphaFold2 model of ARL15. The large asymmetric unit contains four ARL15 and four CNNM2 CBS-pair domains. The relative position of the ARL15 molecules to CBS-pair domains was well defined with 0.6 Å RMSD between the four copies. The protein interfaces were among the best-defined areas in the electron density map. Outside of these sites, there were deviations in the

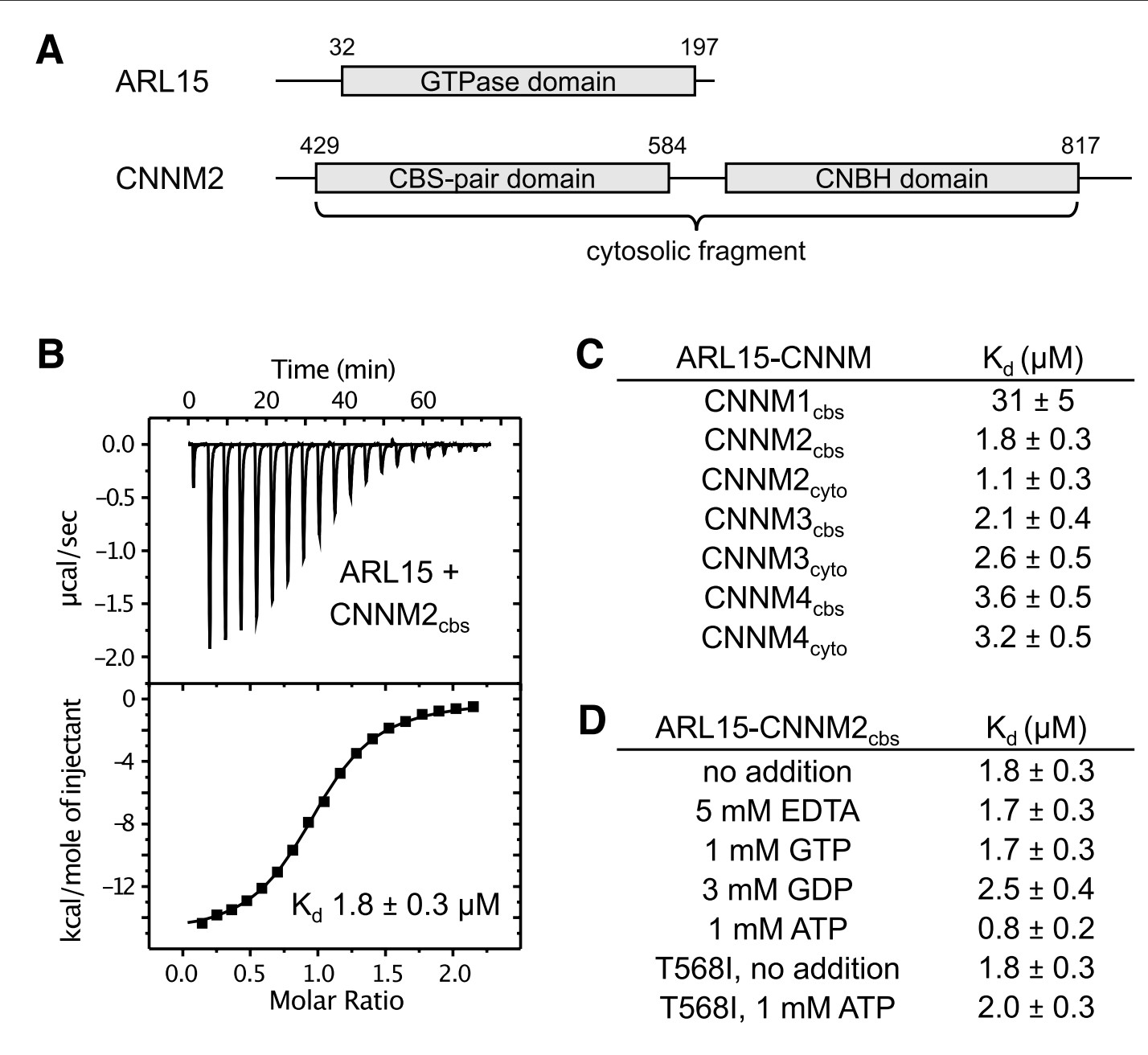

**Figure 2.** ADP-ribosylation factor-like GTPase 15 (ARL15) binds cystathionine-β-synthase-pair domain divalent metal cation transport mediator (CNNM) cystathionine-β-synthase (CBS)-pair domains. (**A**) Domains of ARL15 and CNNM2 proteins used in binding experiments. The four CNNM protein have the same domain organization. (**B**) Isothermal titration calorimetry experiment between the ARL15 GTPase domain and CNNM2 CBS-pair domain. (**C**) Binding affinities ($K_d$) between ARL15 GTPase domain and CBS-pair and cytosolic fragments of CNNM proteins measured by isothermal titration calorimetry (ITC). (**D**) Effect of nucleotides on the affinity of the ARL15-CNNM2 interaction measured by ITC. The CNNM2 T568I mutant, which is unable to bind ATP, bound ARL15 with the same affinity as the wild-type CNNM2 CBS-pair domain but did not show increased affinity upon addition of ATP.

The online version of this article includes the following source data for figure 2:

**Source data 1.** Isothermal titration calorimetry (ITC) thermograms showing binding between ADP-ribosylation factor-like GTPase 15 (ARL15) GTPase domain and cystathionine-β-synthase (CBS)-pair and cytosolic fragments of cystathionine-β-synthase-pair domain divalent metal cation transport mediator (CNNM) proteins in *Figure 1B and C*.

**Source data 2.** Isothermal titration calorimetry (ITC) thermograms showing binding between ADP-ribosylation factor-like GTPase 15 (ARL15) GTPase domain and cystathionine-β-synthase (CBS)-pair domain of CBS-pair domain divalent metal cation transport mediator 2 (CNNM2) in presence of 5 mM EDTA, 1 mM GTP, and 3 mM GDP in *Figure 1D*.

*Figure 2 continued on next page*

Figure 2 continued

**Source data 3.** Isothermal titration calorimetry (ITC) thermograms showing binding between ADP-ribosylation factor-like GTPase 15 (ARL15) GTPase domain and cystathionine-β-synthase (CBS)-pair domain of CBS domain divalent metal cation transport mediator 2 (CNNM2) (wild-type [WT] and T568I mutant) in presence or absence of Mg-ATP in *Figure 1D*.

conformations of some loops and the orientation of the CBS-pair domain N-terminal helix, which are likely due to crystal packing.

## Identification of the ARL15-CNNM interface

As is typical, the CBS-pair domains are present as dimers with a small twist, possibly due to the absence of Mg$^{2+}$-ATP (*Figure 3—figure supplement 1*). This twist generates two contact surfaces between the ARL15 and CNNM2 CBS-pair domains (*Figure 3C*). The larger surface contains numerous polar contacts and involves two ARL15 structural elements: a C-terminal part of helix α2 with following loop and the loop following helix α3 (*Figure 3D*). Helix α2 is a continuation of the Switch II region and is a

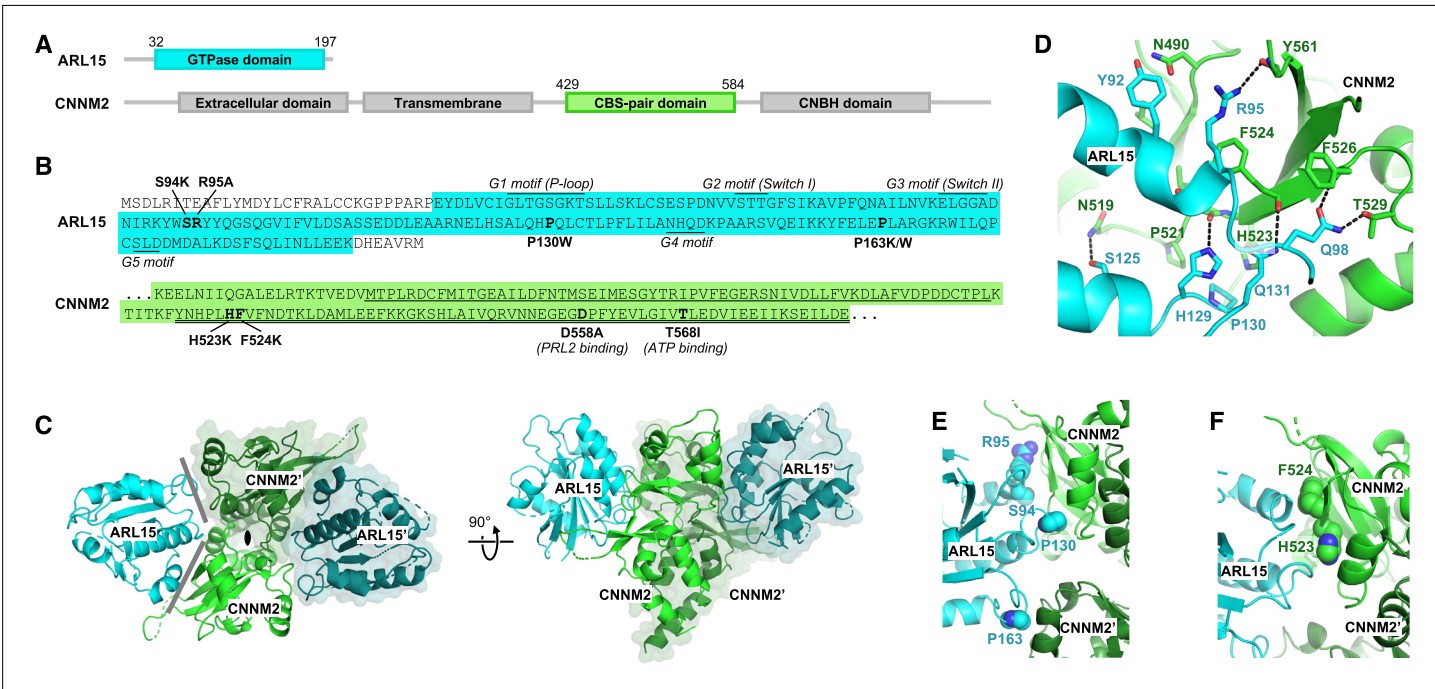

**Figure 3.** Crystal structure of ADP-ribosylation factor-like GTPase 15 (ARL15) bound to the cystathionine-β-synthase (CBS)-pair domain of CBS-pair domain divalent metal cation transport mediator 2 (CNNM2). (**A**) Domain organization of the proteins. (**B**) Sequences of the fragments crystallized and residues mutated. The typical GTPase motifs are indicated for ARL15. Within the CNNM2 CBS-pair domain, the CBS1 and CBS2 motifs are singly and doubly underlined. (**C**) Crystal structure of two ARL15 molecules bound to a CBS-pair domain dimer. There are two distinct interfaces (gray lines) between the ARL15 and CBS-pair monomers. (**D**) Detail of intermolecular contacts between ARL15 and the CNNM2 CBS-pair domain. (**E**) Interfacial ARL15 residues tested by mutagenesis. (**F**) Interfacial CNNM2 residues tested by mutagenesis.

The online version of this article includes the following video and figure supplement(s) for figure 3:

**Figure supplement 1.** Structural comparison of the cystathionine-β-synthase-pair domain divalent metal cation transport mediator 2 (CNNM2) cystathionine-β-synthase (CBS)-pair domain dimerization when (**A**) bound to ADP-ribosylation factor-like GTPase 15 (ARL15) (*green*), (**B**) bound to Mg-ATP (*red, PDB code 4P1O*), (**C**) no ligand bound (*brown, 4IYS*), and (**D**) T568I mutant (*blue, 4IY4*).

**Figure supplement 2.** Comparison of ADP-ribosylation factor-like GTPase 15 (ARL15) to related small GTPases.

**Figure 3—video 1.** Crystal structure of two ADP-ribosylation factor-like GTPase 15 (ARL15) molecules (*cyan* and *teal*) bound to a cystathionine-β-synthase (CBS)-pair domain dimer (*light* and *dark green*).

https://elifesciences.org/articles/86129/figures#fig3video1

**Figure 3—video 2.** Comparison of ADP-ribosylation factor-like GTPase 15 (ARL15) structure (*cyan*) and the related small GTPases: ARL2 (*pink, PDB code 4GOK*), ARL3 (*orange, 3BH6*), ARF1 (*olive, 1O3Y*), ARF6 (*violet, 4KAX*).

https://elifesciences.org/articles/86129/figures#fig3video2

**Table 1.** Statistics of data collection and refinement.

| PDB code | 8F6D |
| --- | --- |
| Data collection | |
| X-ray source | APS 24ID-E |
| Wavelength (Å) | 0.9792 |
| Space group | P1 |
| Cell dimensions | |
| $a$, $b$, $c$ (Å) | 66.02, 73.32, 79.33 |
| $\alpha$, $\beta$, $\gamma$ (°) | 94.4, 95.5, 115.3 |
| Resolution (Å) | 50–3.20 (3.26–3.20)* |
| $R_{merge}$ | 0.078 (0.782) |
| $I/\sigma I$ | 10.4 (0.6) |
| Completeness (%) | 94.2 (93.1) |
| Redundancy | 3.7 (3.8) |
| $CC_{1/2}$ | 0.998 (0.624) |
| Refinement | |
| Resolution (Å) | 29.5–3.20 |
| No. reflections | 20526 |
| $R_{work}/R_{free}$ | 0.264/0.292 |
| No. atoms | |
| Protein | 7791 |
| $B$-factors | |
| Protein | 115.3 |
| R.m.s. deviations | |
| Bond lengths (Å) | 0.004 |
| Bond angles (°) | 0.79 |
| Ramachandran statistics (%) | |
| Most favored regions | 97.98 |
| Additional allowed regions | 1.56 |
| Disallowed regions | 0.46 |

*Highest resolution shell is shown in parentheses.

common binding site for many GTPase effectors. ARL15 Gln98 makes intermolecular hydrogen bonds with CNNM2 Thr529 and the backbone amide of Phe526. ARL15 Gln131 and His129 form hydrogen bonds with CNNM2 backbone carbonyl of Phe524 and Leu522. In one copy of the ARL15-CBS complex, the side chain of Arg95 bonds with the backbone carbonyl of Tyr561. The side chain of Arg95 appears to be partially disordered in other copies which could be a result of competing interactions with other nearby electron donors such as a mainchain carbonyl or the side chain of Asn490. Significantly, all the ARL15-CNNM2 interfaces show hydrophobic stacking between aliphatic part of Arg95 and the side chain of Phe524. Another key hydrophobic binding determinant is provided by ARL15 Pro130 that inserts into a small pocket formed by side chains of CNNM2 Pro521 and His523.

The second binding surface between ARL15 and the CBS-pair domain is smaller and involves ARL15 residues in the loop following helix α4 that sit in a shallow pocket formed by two helices from the CBS-pair domain (*Figure 3D*). These helices are usually part of the CBS-pair dimerization interface

and not exposed in the alternative, flat dimer often observed in crystals of CNNM CBS-pair domains (*Figure 3—figure supplement 1*).

## Structural comparisons of ARL15 to other GTPases

Comparison of the ARL15 structure with other GTPases confirmed it belongs in the ARF/ARL family (*Figure 3—figure supplement 2* and *Figure 3—video 2*). An RMSD of ~1.5 Å was observed between ARL15 and the GTPases ARL2, ARL3, ARF1, and ARF6, reflecting the high degree of sequence identity (around 35%). ARL15 showed a large degree of disorder in the GTP-binding site and loop regions. Although a GTP analog was present in the solution, the binding site was unoccupied, suggesting that nucleotide binding was incompatible with protein-protein contacts in the crystal. The G1 motif (P-loop) was traceable but adopted a conformation incompatible with nucleotide binding while the G2 and G5 motifs were largely disordered.

## Identification of the interaction surface by mutagenesis

We turned to mutagenesis to determine the importance of the two binding surfaces. We mutated Ser94, Arg95, and Pro130 of ARL15 from the larger interaction surface and Pro163 from the smaller interface (*Figure 3*) and tested the mutant proteins for binding to CBS-pair domain of CNNM2 (residues 429–584) using pulldown assays and ITC (*Figure 3E* and *Figure 4*). 1D NMR spectra of the mutants closely matched the spectrum of WT ARL15 confirming that the mutations did not cause protein unfolding (*Figure 4—figure supplement 1*). Pulldowns with GST-fused CNNM2 CBS-pair domain showed binding for WT ARL15 and the P163K mutant but not for S94K, R95A, and P130W (*Figure 4A*). Identical results were obtained in pulldowns using GST-fused CBS-pair domains of CNNM3 and CNNM4 (*Figure 4B–C*). The pulldown results were confirmed by ITC experiments with the CNNM2 CBS-pair domain (*Figure 4D–E*). The R95A and P130W mutations abrogated binding while S94K and S94W decreased the affinity of ARL15 by 20-fold ($K_d$ of 40 µM). On the other hand, mutation of Pro163 at the second interface had no effect. The P163K and P163W mutants showed WT affinity for the CNNM2 CBS-pair domain. These results establish that the first interaction surface is critical for the ARL15-CNNM binding while the second surface is dispensable and likely the result of crystal packing.

We extended these results to the full-length proteins using GST-pulldown and co-immunoprecipitation experiments in HEK-293T cell lysates. The GST-tagged CBS-pair domain of murine CNNM2 efficiently pulled down FLAG-tagged full-length ARL15 (*Figure 4F*). The P163W and P163K mutants showed WT binding, S94W and S94K showed weak binding, and P130W and R95A showed no binding. A reciprocal experiment demonstrated that the same interaction surface is essential for binding full-length CNNM (*Figure 4G*). HEK-293T cells were transfected with WT and R95A mutant FLAG-tagged ARL15 to test binding of endogenous CNNM. Native CNNM3 was efficiently co-immunoprecipitated by FLAG-tagged ARL15 WT but not FLAG-tagged ARL15 mutant (R95A) or in untransfected cells. Thus, Arg95 is essential for the interaction with full-length CNNM3.

## The ARL15-binding site is conserved across the CNNM family

We used the crystal structure to identify point mutations in CNNMs that would knock out ARL15 binding. CNNM2 residues His523 and Phe524 make important contacts with ARL15 in the crystal structure (*Figure 3F* and *Figure 3—video 1*) and are conserved from humans to flies (*Figure 5A*). These residues correspond to His507 and Cys508 in CNNM1, His391, and Phe392 in CNNM3, and H450 and F451 in CNNM4 (*Figure 5B*). As CNNM1 has a cysteine in place of Phe524, we mutated it to phenylalanine to test if that would improve binding. The histidine and phenylalanine residues in the other CNNMs were mutated to lysine as potential loss-of-function mutations. The mutant proteins were prepared, and their proper folding confirmed by NMR spectroscopy (*Figure 5—figure supplement 1*). ARL15 binding was measured in pulldown and ITC experiments (*Figure 5C–H*).

The CNNM1 C508F mutant showed a significantly stronger pulldown of ARL15 than did the WT domain (*Figure 5C*). The increase in affinity was measured by ITC to be 9 µM, a threefold improvement over WT (*Figure 5G*). Interestingly, while the cysteine is perfectly conserved in mammals and most vertebrates, it is replaced by tryptophan in turtles and tortoises. Weaker ARL15 binding appears to be a functional or, at least, a conserved feature of CNNM1 proteins.

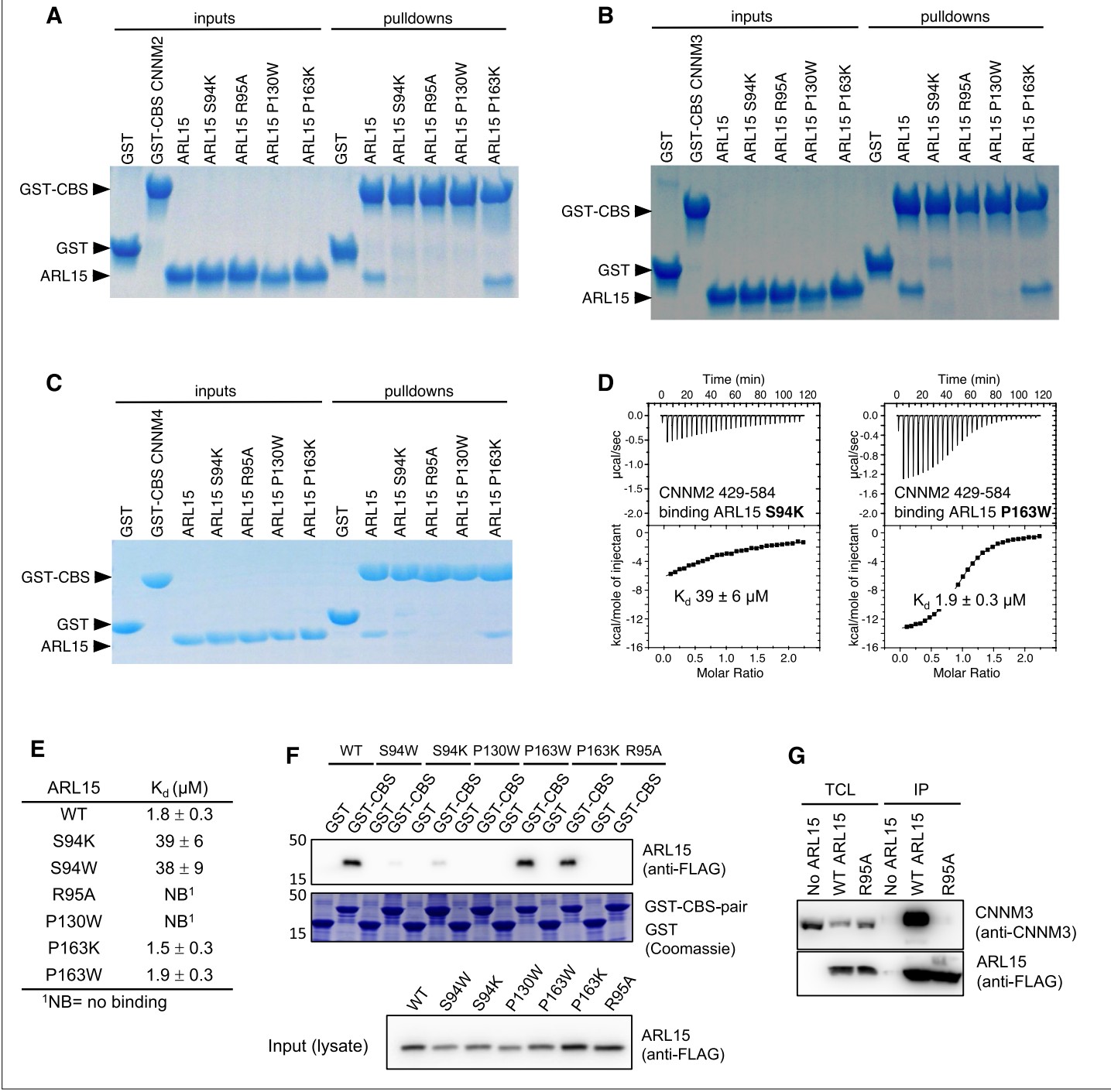

**Figure 4.** Mutagenesis of ADP-ribosylation factor-like GTPase 15 (ARL15) confirms the cystathionine-β-synthase-pair domain divalent metal cation transport mediator (CNNM)-binding site. Pulldowns of recombinant WT and mutant ARL15 GTPase domains (residues 32–197) by (**A**) GST-CNNM2 cystathionine-β-synthase (CBS)-pair domain (residues 429–584), (**B**) GST-CNNM3 CBS-pair domain (residues 299–452), (**C**) GST-CNNM4 CBS-pair domain (residues 356–511). (**D**) Isothermal titration of mutants of ARL15 GTPase domain and CNNM2 CBS-pair domain. (**E**) CNNM2 binding affinities ($K_d$) of ARL15 GTPase mutants measured by isothermal titration calorimetry (ITC). (**F**) Pulldown of wild-type ARL15 (WT) and the indicated mutants expressed in HEK-293T cells by GST-fused CNNM2 CBS-pair domain. (**G**) Co-immunoprecipitation of FLAG-tagged ARL15 with native CNNM3 shows that the ARL15 R95A mutation fully blocks binding. No ARL15, untransfected; WT, wild-type.

The online version of this article includes the following source data and figure supplement(s) for figure 4:

**Source data 1.** Uncropped gels of pulldowns of recombinant wild-type (WT) and mutant ADP-ribosylation factor-like GTPase 15 (ARL15) GTPase domains (residues 32–197) by GST-cystathionine-β-synthase-pair domain divalent metal cation transport mediator 2 (CNNM2) cystathionine-β-synthase

*Figure 4 continued on next page*

*Figure 4 continued*

(CBS)-pair domain (residues 429–584) in *Figure 4A–C*.

**Source data 2.** Isothermal titration calorimetry (ITC) thermograms of cystathionine-β-synthase-pair domain divalent metal cation transport mediator 2 (CNNM2) cystathionine-β-synthase (CBS)-pair domain binding mutants of ADP-ribosylation factor-like GTPase 15 (ARL15) GTPase domain (S94K, S94W, R95A, P130W, P163K, and P163W) in *Figure 4D and E*.

**Source data 3.** Uncropped blots and gels of pulldown of wild-type (WT) ADP-ribosylation factor-like GTPase 15 (ARL15) (WT) and the indicated mutants expressed in HEK-293T cells by GST-fused cystathionine-β-synthase-pair domain divalent metal cation transport mediator 2 (CNNM2) cystathionine-β-synthase (CBS)-pair domain in *Figure 4F*.

**Source data 4.** Uncropped blots images of co-immunoprecipitation of FLAG-tagged ADP-ribosylation factor-like GTPase 15 (ARL15) with native cystathionine-β-synthase-pair domain divalent metal cation transport mediator 3 (CNNM3) show the R95A mutation fully blocks binding in *Figure 4G*.

**Figure supplement 1.** ADP-ribosylation factor-like GTPase 15 (ARL15) mutants are properly folded.

Mutation of either the phenylalanine or histidine residues caused a complete loss of ARL15 binding to either the CBS-pair domains or larger cytosolic fragments including the CNBH domain in pulldown assays (*Figure 5D–F*). This was largely confirmed by ITC titrations, which showed complete loss of binding for the histidine mutants (*Figure 5H*). The phenylalanine mutants showed small enthalpic signals in the ITC thermograms, suggesting very weak binding, but the affinities were too weak to be quantified. In summary, the mutagenesis results corroborate the binding surfaces observed in the crystal structure and the concurrence of the results across the four CNNM isoforms confirms that ARL15 binds identically to all the proteins.

## ARL15 and PRLs compete for binding

Both ARL15 and PRLs bind CNNM; therefore, it is relevant to ask if they compete for binding. Structural superposition of the complexes with the CNNM2 CBS-pair domain shows considerable overlap in positions of ARL15 and PRL1 (*Figure 6A* and *Figure 6—video 1*). This strongly suggests they cannot bind at the same time. We tested this using ITC to look for competitive binding. PRLs bind to CNNM with approximately 100-fold better affinity and would be expected to outcompete ARL15 for binding. Indeed, titration of ARL15 into a mixture of CNNM2 CBS and PRL2 resulted in essentially no heat released indicating no binding occurred (*Figure 6B*). As independent verification, we carried out a competition experiment using NMR (*Figure 6C*). Addition of the CNNM2 CBS-pair domain (429–584) to $^{15}$N-labeled ARL15 (32–197) resulted in the disappearance of many signals in the $^1$H-$^{15}$N correlation spectrum due to the formation of a higher molecular weight complex. In the spectrum with both PRL2 and the CBS-pair domain, all the NMR signals were visible (*Figure 6C*), thus showing that PRL2 prevented the CBS-pair domain binding to ARL15.

We repeated these experiments with the CNNM2 T568I mutant that has been suggested to lock the CBS-pair dimer into a flat conformation (*Corral-Rodríguez et al., 2014*). The results of the ITC and NMR titrations were the same with WT CNNM2 confirming that ARL15 and PRL2 compete for binding independently of the conformation of the dimer (*Figure 6—figure supplement 1*).

## Partner specific CNNM mutants

Analysis of the structures suggested that it should be possible to design CNNM mutants to specifically block binding of ARL15 or PRLs. It was already known that CNNM proteins bind PRLs through an aspartic acid that acts as a substrate mimic (*Gulerez et al., 2016*; *Zhang et al., 2017*; *Giménez-Mascarell et al., 2017*). We mutated that aspartic acid (Asp558) in CNNM2 and assessed its ability to bind ARL15 and PRL2 by ITC. We also tested if the CNNM2 H523K mutant was affected in its ability to bind PRL2 (*Figure 6D–E*).

The mutants showed completely complementary effects. The D558A mutant bound to ARL15 with the same affinity as WT CBS-pair domain but was unable to interact with PRL2. Conversely, the H523K mutant bound PRL2 with nanomolar affinity but completely blocked ARL15 binding. The existence of mutants that specifically impair ARL15 versus PRL binding should be useful in future biological studies.

## ARL15 inhibits CNNM activity

CNNM transporter activity can be measured in cells using a fluorescent indicator, Magnesium Green, that measures cytosolic Mg$^{2+}$ levels (*Yamazaki et al., 2013*). Cells overexpressing CNNMs show Mg$^{2+}$

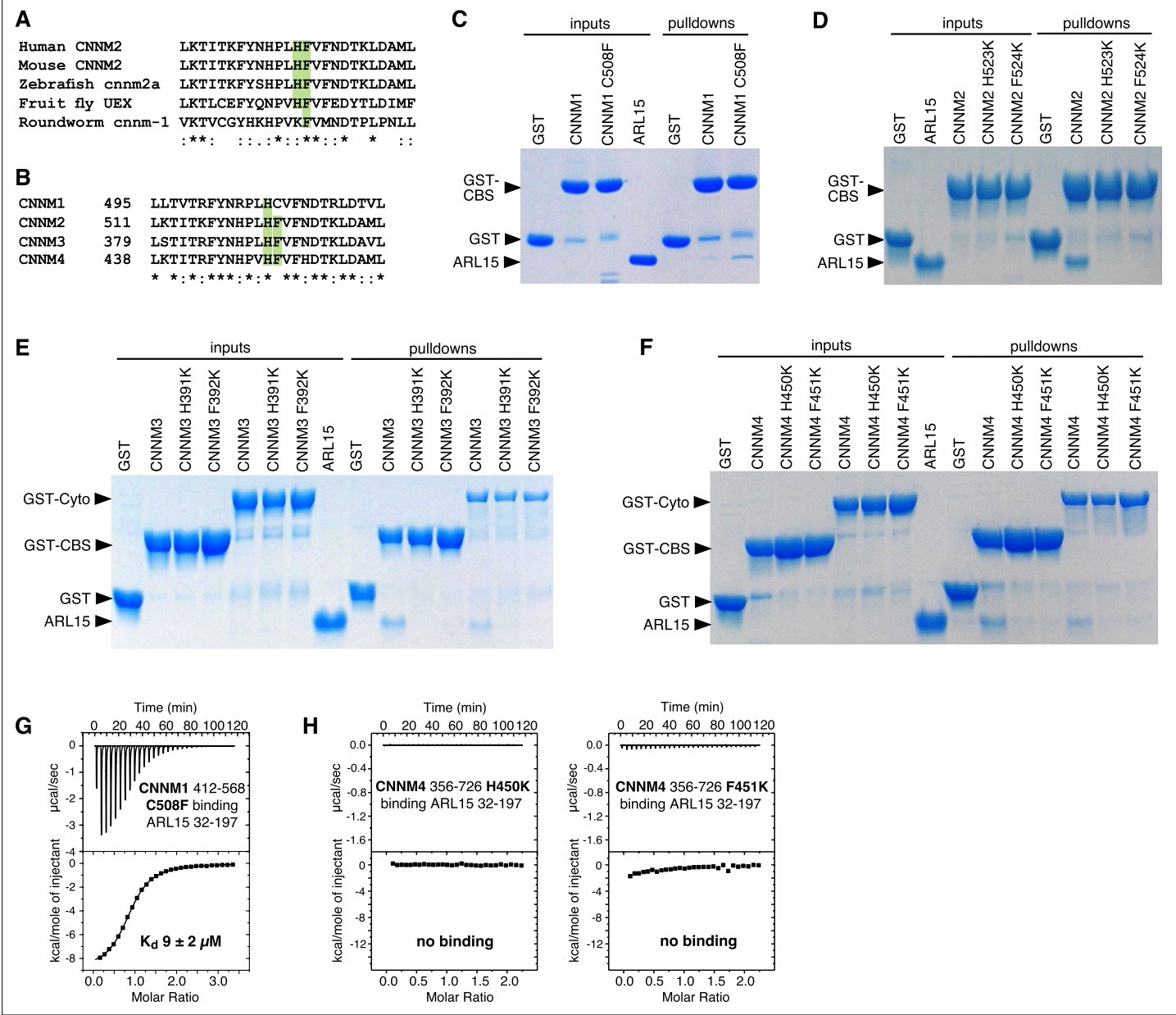

**Figure 5.** Mutagenesis of cystathionine-β-synthase-pair domain divalent metal cation transport mediator (CNNM) confirms conservation of ADP-ribosylation factor-like GTPase 15 (ARL15)-binding site. (**A**) Conservation of key ARL15-binding residues (*green*) in different species. (**B**) Conservation of ARL15-binding residues in human CNNMs. (**C–F**) Pulldown of recombinant GTPase domain (residues 32–197) by GST fusions with (**C**) CNNM1 (residues 412–568), (**D**) CNNM2 (residues 429–584), (**E**) CNNM3 (residues 299–452 and residues 299–658), (**F**) CNNM4 (residues 356–511 and residues 356–726). (**G**) Isothermal titration calorimetry (ITC) thermogram of ARL15 GTPase domain and CNNM1 cystathionine-β-synthase (CBS)-pair domain with C508F mutation shows a threefold improvement in affinity. (**H**) Thermograms of CNNM4 cytosolic fragment H450K and F451K mutants show no ARL15 binding.

The online version of this article includes the following source data and figure supplement(s) for figure 5:

**Source data 1.** Uncropped gels of pulldown of recombinant GTPase domain (residues 32–197) by wild-type (WT) and mutant GST-fused cystathionine-β-synthase-pair domain divalent metal cation transport mediator (CNNM) 1–4 in *Figure 5C–F*.

**Source data 2.** Isothermal titration calorimetry (ITC) thermograms of ADP-ribosylation factor-like GTPase 15 (ARL15) GTPase domain binding mutants of cystathionine-β-synthase-pair domain divalent metal cation transport mediator (CNNM) 1–4 cystathionine-β-synthase (CBS)-pair domains and CNNM4 cytosolic domain in *Figure 5G and H*.

**Figure supplement 1.** Cystathionine-β-synthase-pair domain divalent metal cation transport mediator 2 (CNNM2) cystathionine-β-synthase (CBS)-pair domain mutants are properly folded.

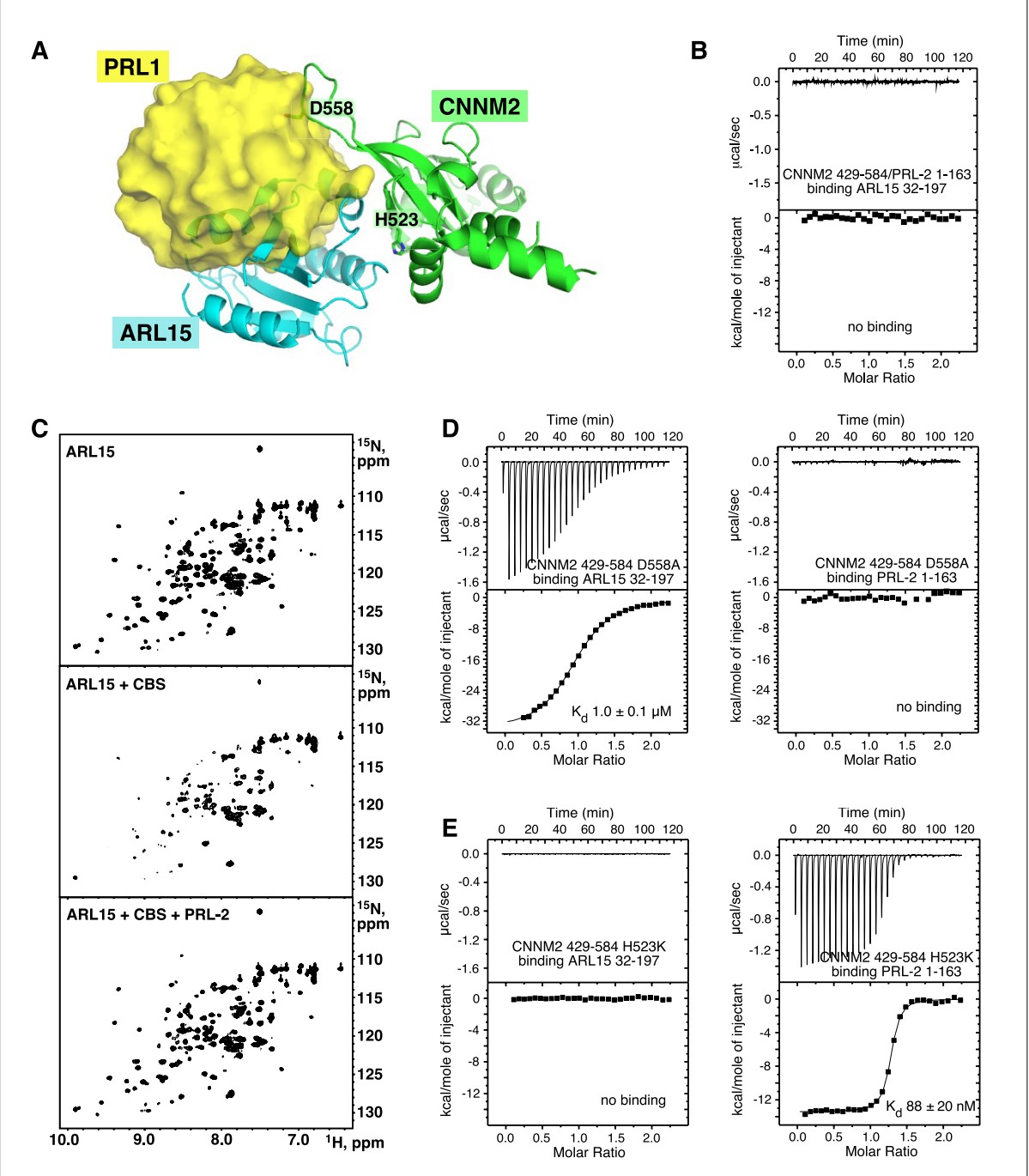

**Figure 6.** ADP-ribosylation factor-like GTPase 15 (ARL15) and phosphatases of regenerating liver (PRLs) have overlapping but distinct binding sites. (**A**) Overlay of the structures of the cystathionine-β-synthase-pair domain divalent metal cation transport mediator 2 (CNNM2) complexes with ARL15 (cyan) and PRL1 (yellow, PDB code 6WUS) shows that simultaneous binding is not possible. (**B**) Isothermal titration calorimetry (ITC) thermogram shows no binding of ARL15 to the preformed complex of CNNM2 and PRL2. (**C**) ¹H-¹⁵N correlation NMR spectra of 100 μM ¹⁵N-labeled ARL15 alone (top), in the presence of 100 μM CNNM2 cystathionine-β-synthase (CBS)-pair domain (middle), and after addition of 150 μM PRL2 (bottom). The ARL15 NMR signals are attenuated upon binding CNNM2 but reappear when ARL15 is displaced by PRL2. (**D**) ITC thermograms demonstrating that the CNNM2 D558A mutation specifically disrupts PRL2 binding. (**E**) ITC thermograms demonstrating that the CNNM2 H523K mutation specifically disrupts ARL15 binding.

The online version of this article includes the following video, source data, and figure supplement(s) for figure 6:

**Source data 1.** Isothermal titration calorimetry (ITC) experiments showing ADP-ribosylation factor-like GTPase 15 (ARL15) and phosphatases of regenerating liver (PRLs) have overlapping but distinct cystathionine-β-synthase-pair domain divalent metal cation transport mediator 2 (CNNM2)-

*Figure 6 continued on next page*

*Figure 6 continued*

binding sites in *Figure 6B, D, and E*.

**Figure supplement 1.** Competition between ADP-ribosylation factor-like GTPase 15 (ARL15) and phosphatase of regenerating liver 2 (PRL2) binding to cystathionine-β-synthase-pair domain divalent metal cation transport mediator 2 (CNNM2) is not affected by the CNNM2 T568I mutation.

**Figure 6—video 1.** Superposition of crystal structures of cystathionine-β-synthase-pair domain divalent metal cation transport mediator 2 (CNNM2) cystathionine-β-synthase (CBS)-pair domain (*green*) bound to ADP-ribosylation factor-like GTPase 15 (ARL15) (*cyan*) and phosphatase of regenerating liver 1 (PRL1) (*yellow, PDB code 6WUS*).

efflux in cells when in Mg²⁺-free solution. PRL3 inhibits CNNM4 activity by binding to the CBS-pair domain (*Funato et al., 2014*; *Gulerez et al., 2016*; *Kozlov et al., 2020*). We took advantage of this assay to test if ARL15 affects CNNM activity.

HEK-293T cells were transfected with a mouse CNNM2 mCherry fusion protein with or without co-transfection with FLAG-ARL15 (*Figure 7A*). CNNM2-expressing cells showed a 35% drop in Mg²⁺ levels, while untransfected cells showed little or no drop. Cells expressing both CNNM2 and ARL15 showed no efflux, thus demonstrating that, like PRL3, ARL15 inhibits CNNM protein activity. The

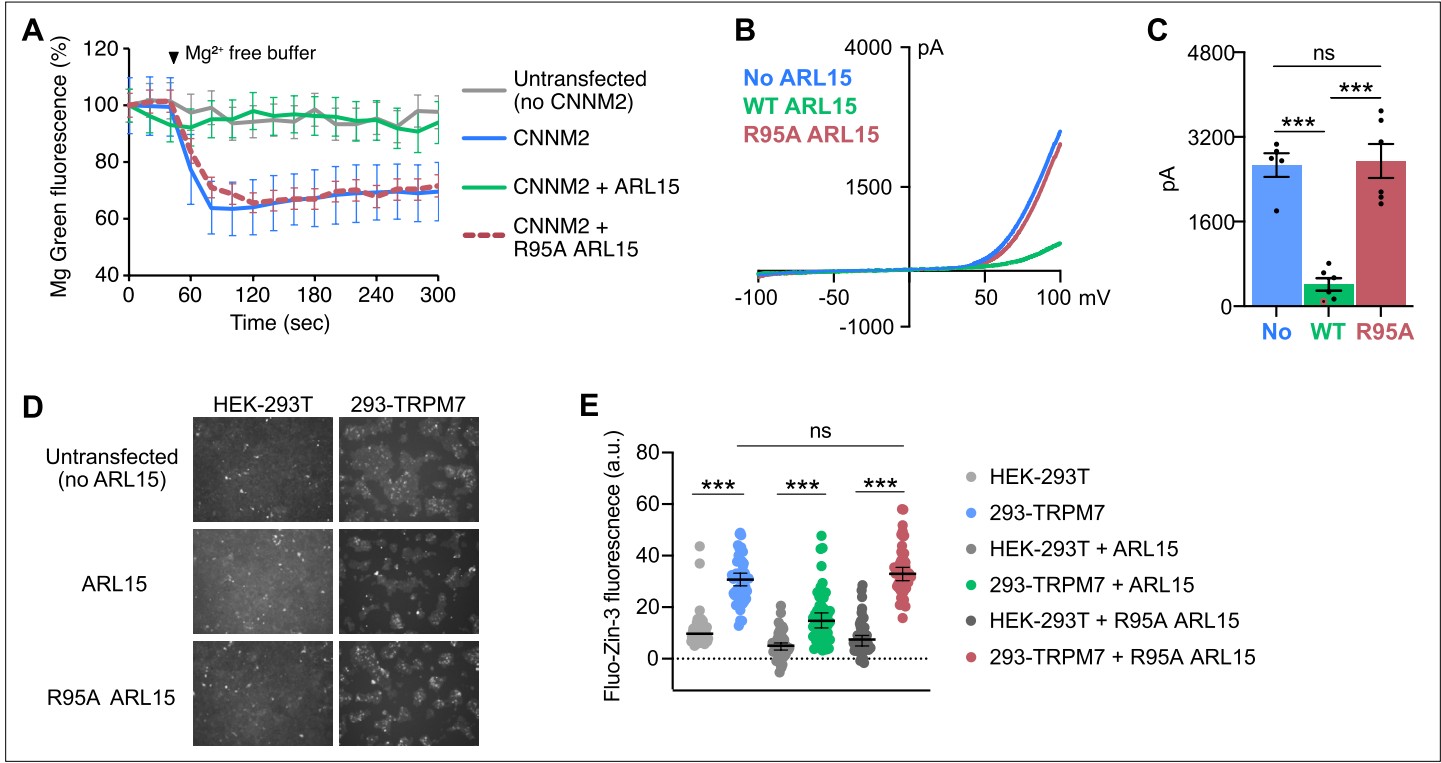

**Figure 7.** Disruption of the cystathionine-β-synthase-pair domain divalent metal cation transport mediator (CNNM)-binding site blocks ADP-ribosylation factor-like GTPase 15 (ARL15) regulation of CNNM and transient receptor potential ion channel subfamily M member 7 (TRPM7) divalent cation transport. (**A**) CNNM2 Mg²⁺ efflux is inhibited by ARL15 but not the R95A mutant. HEK-293T cells transfected with the indicated constructs were loaded with a magnesium sensitive dye and fluorescence monitored following Mg²⁺ depletion. The mean relative fluorescence intensities of 10 cells are shown. (**B**) Representative TRPM7 whole-cell currents from 293-TRPM7 cells from untransfected controls (no ARL15), wild-type, and R95A mutant ARL15 transfectants. (**C**) Average current density of the different groups from (**B**). (**D**) Zinc influx assay using the FluoZin-3 Zn²⁺ indicator was used to monitor TRPM7 function in untransfected cells or transfected with wild-type or mutant ARL15. Images shown are taken at a time point 5–10 min after application of 30 µM ZnCl₂ to stimulate Zn²⁺ influx. (**E**) Cell counts of fluorescence intensity from (**D**). A total of 50 cells per condition were randomly selected for quantification. *** indicates a p-value of less than 0.0001.

The online version of this article includes the following source data and figure supplement(s) for figure 7:

**Source data 1.** Statistical analysis of the decrease in fluorescence in *Figure 7A*.

**Figure supplement 1.** Localization of HEK239T cells transfected with mCNNM2-mCherry and FLAG-ADP-ribosylation factor-like GTPase 15 (ARL15).

ARL15 R95A mutant was unable to inhibit CNNM2 function confirming that the inhibition is due to ARL15 binding to the CBS-pair domain.

As ARL15 has been reported to affect CNNM glycosylation (*Zolotarov et al., 2021*), we used confocal microscopy to verify that the ARL15 did not alter the levels of plasma membrane expression of CNNM2 (*Figure 7—figure supplement 1*). Both CNNM2 and ARL15 showed mixed localization on the plasma membrane and internal membranes, possibly the Golgi apparatus. Imaging of the ARL15 R95A mutant further confirmed that the mutation did not disrupt the plasma membrane localization of ARL15.

## CNNM-binding-defective ARL15 is unable to inhibit TRPM7 channel activity

We carried out related experiments to assess the effect of the ARL15-CNNM interaction on TRPM7 function. Overexpression of ARL15 has been shown to inhibit $Zn^{2+}$ influx in cells by TRPM7 (*Bai et al., 2021*; *Kollewe et al., 2021*). We compared the effects of transfection of WT ARL15 and the R95A mutant in two assays of TRPM7 channel function. Whole-cell electrophysiological recordings of cells with and without co-expression of ARL15 confirmed that ARL15 suppresses ion influx (*Figure 7B–C*). In contrast, co-expression of the ARL15 R95A mutant showed no suppression with currents indistinguishable from the untransfected cells. Similar results were observed with a $Zn^{2+}$ influx assay, taking advantage of TRPM7 permeability to $Zn^{2+}$ (*Monteilh-Zoller et al., 2003*), comparing cells with (293-TRPM7) and without (HEK-293T) expression of TRPM7 (*Figure 7D–E*). ARL15 co-expression markedly decreased intracellular $Zn^{2+}$ levels in TRPM7-expressing cells while expression of the ARL15 R95A mutant had no effect.

## Discussion

### ARL15 is an atypical GTPase

Our observation of very weak binding of GTP to ARL15 and low intrinsic GTPase activity conflicts with a recent report that ARL15 upregulates TGFβ family signaling by promoting the assembly of the Smad complex (*Shi et al., 2022*). That study observed that the MH2 domain of Smad4 binds to ARL15 and acts as a GTPase activating protein (GAP). While the discrepancy could be due to the absence of ARL15 palmitoylation in our biochemical studies, we feel a more likely explanation is that ARL15 is an atypical GTPase. Notably, ARL15 has an alanine at position 86 (*Figure 3—figure supplement 2A*). In almost all other ARLs, this position is occupied by glutamine, which plays a role in GTP hydrolysis. The residue is often mutated to leucine to block GTPase activity (*Sprang, 1997*; *Sztul et al., 2019*) and, in fact, Shi et al. used the A68L mutation to stabilize the Smad4-ARL15 interaction (*Shi et al., 2022*). It is questionable whether ARL15 has catalytic activity, given the importance and prevalence of glutamine in related GTPases.

Weak affinity of ARL15 for GTP is a property shared with several other ARL-family members. ARL2, ARL3, and ARL13B have all been reported to bind guanine nucleotides with micromolar affinity (*Gulerez et al., 2016*; *Linari et al., 1999*; *Ivanova et al., 2017*). We see rapid binding of GTP to ARL15 in ITC experiments (*Figure 1D*), so it is unlikely a GEF is required to facilitate GTP binding. It is impossible to affirm that a GAP does not exist for ARL15, but this seems unlikely since, in presence of rapid exchange, ARL15 GTPase activity would constitute an unregulated, futile cycle depleting the cell of GTP.

A second notable sequence difference between ARL15 and other small GTPases is the connecting region between Switch I and Switch II. This region is characteristic of ARF/ARL GTPase family members and contains two β-strands and a patch of three conserved aromatic residues that participate in the interactions with effectors (*Ménétrey et al., 2007*). In ARL15, the three residues are Phe66, Lys81, and Tyr96. Lys81 replaces a tryptophan found in all ARFs and many ARLs, which suggests that ARL15 interacts with effectors differently from other ARF/ARL GTPases.

### Complex with CNNM

CBS-pair domains, also termed Bateman domains, are found in many proteins including a chloride channel ClC and bacterial $Mg^{2+}$ transporter MgtE (*Bateman, 1997*; *Baykov et al., 2011*). They consist of repeated CBS motifs - CBS1 and CBS2 - that fold together to form a $Mg^{2+}$-ATP binding site

(*Ereño-Orbea et al., 2013*). The CNNM CBS-pair domains most often dimerize in a head-to-head manner forming a ring around two ATP molecules but considerable plasticity exists at the dimerization interface with many different conformations observed in crystal structures (*Corral-Rodríguez et al., 2014*; *Giménez-Mascarell et al., 2017*; *Gulerez et al., 2016*; *Hanzal-Bayer et al., 2005*; *Zhang et al., 2017*; *Chen et al., 2021*; *Chen et al., 2020*; *Fakih et al., 2023*). Zolotarov and colleagues had previously proposed an in silico model of ARL15 bound to CNNM2 (*Zolotarov et al., 2021*). That model posited that the first CBS motif and CNBH domain of CNNM2 bind ARL15. This is inconsistent with our observations that residues in the second CBS motif are essential for binding while the CNBH domain is dispensable (*Figure 4*). The model also differed in the prediction of the ARL15-binding surface.

More recent work from the same group reported that PRL and ARL15 compete for binding to CNNMs in agreement with our results (*Hardy et al., 2023*). PRL-2 overexpression decreased ARL15 binding to CNNM3 and enhanced the function of TRPM7. In cells, competition between the two proteins will be affected by relative abundance of ARL15 and PRLs and their post-translational modifications. Based on its ~100-fold higher affinity in vitro, PRLs should outcompete ARL15 for CNNM binding but this would depend on the local protein concentrations. ARL15 is dynamically and reversibly membrane associated due to its palmitoylation (*Wu et al., 2021*). PRLs are similarly membrane associated due to C-terminal farnesylation (*Cates et al., 1996*; *Zeng et al., 2000*; *Wang et al., 2002*). PRLs undergo additional modifications of their catalytic cysteine which regulate their affinity for binding CNNM proteins (*Gulerez et al., 2016*). The dimeric nature of CNNM proteins adds an additional complication as ARL15 and a PRL could bind simultaneously to separate monomers of a CNNM dimer.

## Cellular function

Given that ARL15 and PRLs bind to overlapping sites on CNNMs, it is perhaps not surprising that ARL15 inhibits CNNM2 $Mg^{2+}$ efflux. PRLs inhibit $Mg^{2+}$ efflux by CNNM4 through direct binding to the CBS-pair domain. That ARL15 regulation of TRPM7 occurs through ARL15-CNNM binding is more surprising. It is possible that the R95A mutation does more than disrupt CNNM binding but the simplest model is that ARL15 and PRLs regulate divalent cation transport by binding to a complex of CNNM and TRPM7 (*Figure 8*). While speculative, the model is useful as it recapitulates several observations. Among these are: (1) CNNMs mediate $Mg^{2+}$ efflux (*Yamazaki et al., 2013*; *Chen et al., 2021*; *Huang et al., 2021*), (2) PRLs inhibit $Mg^{2+}$ efflux by CNNMs (*Funato et al., 2014*; *Gulerez et al., 2016*; *Kozlov et al., 2020*), (3) CNNMs and PRLs stimulate divalent cation uptake by TRPM7 (*Bai et al., 2021*; *Kollewe et al., 2021*; *Hardy et al., 2023*), (4) ARL15 forms a complex with TRPM7 and CNNMs and inhibits both proteins (*Kollewe et al., 2021*; *Zolotarov et al., 2021*; *Figures 3 and 7*). The model highlights the importance of coordinated regulation of cation transport. CNNMs are electroneutral ion antiporters (*Chen et al., 2021*; *Yamazaki et al., 2013*) while TRPM7 is electrogenic (*Monteilh-Zoller et al., 2003*). Activating both transporters simultaneously would constitute a futile $Na^+$ leak

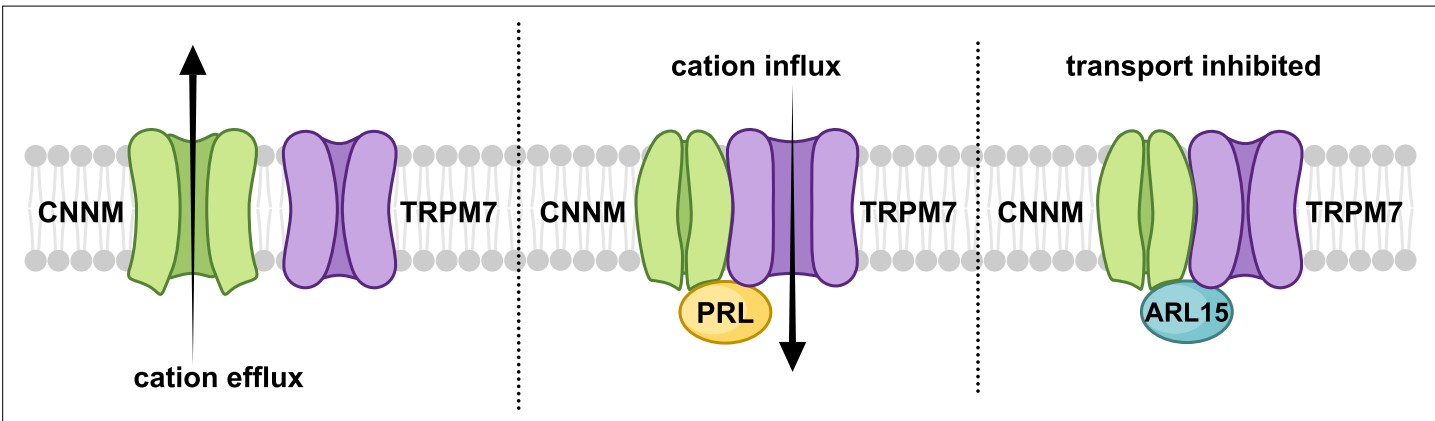

**Figure 8.** Model of regulation of cystathionine-β-synthase-pair domain divalent metal cation transport mediator (CNNM)-transient receptor potential ion channel subfamily M member 7 (TRPM7) complex by ADP-ribosylation factor-like GTPase 15 (ARL15) and phosphatases of regenerating liver (PRLs). CNNM on their own mediate cation efflux. PRL binding to CNNM inhibits cation efflux via CNNMs and activates TRPM7 influx. ARL15 binding to CNNM inhibits both TRPM7 and CNNM transport activity.

and likely be deleterious. Looking forward, many questions remain about the function of CNNMs and TRPM7 on internal membranes, the role of post-translational modifications and membrane trafficking in their regulation, and, from a structural point-of-view, how the ternary complex of monomeric (ARL15), dimeric (CNNM), and tetrameric (TRPM7) proteins is assembled.

## Materials and methods

### Expression and purification of recombinant proteins

A plasmid for bacterial expression of human ARL15 (1–204) was obtained by cloning codon-optimized synthetic DNA into the NdeI and BamHI cloning sites of pET15b (Bio Basic). A truncated construct (residues 32–197) and point mutants were obtained by mutagenesis using the QuikChange Lightning Site-Directed Mutagenesis Kit (Thermo Fisher Scientific). For crystallization, the N-terminal His-tag was deleted and the HHHHH sequence was inserted after residue 197 followed by a stop codon. For HEK-293T expression, human ARL15 (NM_019087.3) in the pcDNA3.1+/C-(K)-DYK vector (GenScript) was purchased and further modified to introduce S94W, S94K, P130W, P163W, P163K, and R95A mutations. The plasmid expressing mouse CNNM2 (residues 1–875)-mCherry-HA was prepared by cloning mCherry-HA into the NotI and XbaI cloning sites of pEG BacMam vector (from Eric Gouaux) and then cloning mouse CNNM2 (residues 1–875) into SalI and NotI cloning sites. Plasmids expressing human CNNM1 (residues 412–568), CNNM2 (residues 429–584 and 429–817), CNNM3 (residues 299–452 and 299–658), CNNM4 (residues 356–511 and 356–726) were described previously (*Chen et al., 2020*). The point mutants of CNNMs were obtained by mutagenesis using the QuikChange Lightning Site-Directed Mutagenesis Kit (Thermo Fisher Scientific). The amino acid sequence of the CBS-pair domain of murine CNNM2 (495–582) was codon optimized for bacterial expression and subcloned into the EcoRI/XhoI sites of pGEX-6P3 vector (GenScript). The plasmid expressing His-tagged human PRL2 (residues 1–163) was described previously (*Gulerez et al., 2016*). DNA sequencing was used to verify all sequence modifications.

Proteins were expressed in *E. coli* BL21(DE3) at 37°C in Luria Broth. ARL15 expression was induced with 0.5 mM IPTG for 4 hr at 30°C. CNNM cytosolic fragments were induced with 0.5 mM IPTG overnight at 18°C, while CBS-pair domains were induced with 0.5 mM IPTG for 4 hr at 30°C. For NMR experiments, the recombinant protein was isotopically labeled by growth of *E. coli* BL21 in M9 minimal medium with $^{15}$N-ammonium sulfate as the sole source of nitrogen.

For His-tagged protein purification, cells were harvested and broken in lysis buffer (50 mM HEPES pH 7.6, 0.5 M NaCl, 5% glycerol) containing 1 mM PMSF, 0.1 mg/ml lysozyme, 0.04% β-mercaptoethanol. His-tagged proteins were purified by affinity chromatography on Ni-NTA agarose resin (QIAGEN) and eluted with buffer containing 0.5 M imidazole. For GST-tagged protein purification, cells were harvested and broken in 1× PBS (phosphate buffered saline) containing 1 mM PMSF, 0.1 mg/ml lysozyme, 0.04% β-mercaptoethanol. GST-tagged proteins were purified on Glutathione Sepharose resin (GE Healthcare) and eluted with buffer containing 20 mM reduced glutathione. The GST-tag was removed by overnight incubation with PreScission Protease, leaving an N-terminal Gly-Pro-Leu-Gly-Ser extension. For final purification, the proteins were applied to a HiLoad 16/600 Superdex 75 or 200 size exclusion column (Cytiva) in HPLC buffer (50 mM HEPES pH 7.5, 200 mM NaCl, 1 mM MgCl$_2$, 1 mM tris(2-carboxyethyl)phosphine hydrochloride [TCEP-HCl]). PRL2 (1–163) was purified using HPLC buffer with higher concentration of the reducing agent (50 mM HEPES pH 7.5, 200 mM NaCl, 1 mM MgCl$_2$, 5 mM TCEP-HCl). The final purified proteins were concentrated to around 10–25 mg/ml (estimated by UV absorbance), and the purity was verified by SDS-PAGE.

### NMR spectroscopy

NMR spectra were acquired at 25°C in 50 mM HEPES pH 7.5, 200 mM NaCl, 1 mM MgCl$_2$, 1 mM TCEP-HCl. In experiments with PRL2, the TCEP concentration was raised to 5 mM to keep the phosphatase catalytic cysteine fully reduced. Phosphorus NMR spectra were acquired on a 11.7T Bruker spectrometer with a 67° flip angle, 6 s between scans, and proton decoupling. Proton-detected NMR experiments were acquired on a 14.1T Bruker 600 MHz spectrometer with cryoprobe. $^{1}$H-$^{15}$N correlation spectra were processed using NMRPipe (*Delaglio et al., 1995*) and analyzed with SPARKY (*Goddard and Kneller, 2008*).

## Isothermal titration calorimetry

ITC experiments were performed on MicroCal VP-ITC titration calorimeter (Malvern Instruments Ltd). The syringe was typically loaded with 150 or 300 µM concentration of the ligand, while the sample cell contained 15 or 30 µM protein. In selected experiments, ARL15 was preloaded with nucleotide and excess nucleotide removed on a desalting column. All experiments were carried out at 20°C with 19 injections of 15 µl or 29 injections of 10 µl. Results were analyzed using ORIGIN software (MicroCal) and fitted to a binding model with a single set of identical sites.

## Crystallization

Initial crystallization conditions were identified utilizing hanging drop vapor diffusion with the Classics II and ProComplex screens (QIAGEN). The best crystals were obtained by equilibrating a 0.6 µl drop at 12 mg/ml of the complex of ARL15 (32–197) and CNNM2 CBS domain (429–584) with close to a stoichiometric amount of GppNHp, non-hydrolyzable GTP analog (Abcam) in HPLC buffer mixed with 0.6 µl of reservoir solution containing 0.2 M sodium chloride, 0.1 M Tris pH 8.5, 25% (wt/vol) PEG3350. Crystals grew in 1–2 days at 20°C. For data collection, crystals were cryo-protected by soaking in the reservoir solution supplemented with 30% (vol/vol) ethylene glycol.

## Structure solution and refinement

Diffraction data from single crystals of ARL15 (32–197)-CNNM2 CBS complex were collected at the Advanced Photon Source (APS) (*Table 1*). Data processing and scaling were performed with DIALS (*Otwinowski and Minor, 1997*). The initial phases for the complex structure were determined by molecular replacement with Phaser (*McCoy et al., 2007*), using the coordinates of the CNNM2 CBS domain (PDB entry 4IY0) (*Giménez-Mascarell et al., 2017*) and an AlphaFold2 model of ARL15 (*Jumper et al., 2021*). The initial phases were improved by Autobuilder in PHENIX package (*Adams et al., 2010*). The starting protein model was then completed and adjusted with the program Coot (*Emsley and Cowtan, 2004*) and improved by multiple cycles of refinement, using the program phenix.refine (*Adams et al., 2010*) and model refitting. At the latest stage of refinement for both structures, we also applied the translation-libration-screw (TLS) option (*Winn et al., 2003*). The final models have 99.5% residues in the allowed regions of Ramachandran plot. The coordinates have been deposited with the Protein Data Bank (PDB) under the accession number 8F6D. Refinement statistics are given in *Table 1*.

## GST-pulldown assays

For GST-pulldown with purified recombinant proteins, a slurry of 25 µl of Glutathione Sepharose beads (Cytiva Sweden AB) was washed twice with 1 ml of pulldown buffer (50 mM HEPES pH 7.5, 200 mM NaCl, 1 mM MgCl$_2$, 1 mM TCEP-HCl, 0.02% Igepal). In between the washes, beads were sedimented by centrifuging at 13,000 rpm for 1 min at 4°C and supernatant was discarded. Two hundred µl of 1 mg/ml GST-fused protein was added to the beads and incubated on ice for 15 min. The beads were washed twice again with 1 ml of buffer as mentioned previously. Fifty µl of the binding partner (1 mg/ml) was added to the beads and incubated on ice for 30 min. On washing the beads thrice with 200 µl of buffer, the proteins were eluted with 25 µl of 20 mM reduced glutathione solution. 20 µl of the eluate was transferred and mixed with 5 µl of 5× SDS loading dye. 15 µl of the sample was loaded onto the gels. Pulldown results were verified by SDS-PAGE. For GST-pulldown of ARL15 and mutants expressed in HEK-293T cells, GST and GST-fused CBS-pair domain of murine CNNM2 were expressed in transformed BL21 (DE3) cells (Stratagene, CA, USA). Bacterial cells were lysed by sonication in ice-cold PBS containing 1% Triton X-100 and protease inhibitor phenylmethylsulfonyl fluoride (Sigma-Aldrich). The bacterial cell lysates were then incubated with glutathione agarose (Sigma-Aldrich) overnight at 4°C with rotation. The agarose beads were washed with PBS with 1% Triton X-100. Concentrations of the GST-fused proteins were measured by Coomassie stain on the SDS-PAGE gel using serial diluted BSA proteins as controls. To express ARL15 proteins, 10 µg of the WT and mutant ARL15 plasmids were transiently transfected into HEK-293T cells plated in a 10 cm dish. After 24 hr, the cells were lysed in 1 ml of mild lysis buffer containing protease inhibitor mixture (Roche Life Sciences) and phosphatase inhibitor mixture (EMD Millipore), and spun down at 14,000×*g* for 10 min at 4°C. For GST-pulldown assays, the cell lysate supernatants were incubated with glutathione agarose bound with 20 µg of GST or GST-fused CBS-pair domain proteins overnight at 4°C

with rotation. The bound proteins were washed with 1 ml of PBS with 0.1% Triton X-100 three times by rotation and eluted into 50 µl of SDS sample buffer. Lysates input (20 µl) and pulldown samples were separated on SDS-PAGE gels and analyzed by immunoblotting. The rabbit monoclonal FLAG antibody (#14793, Cell Signaling Technology) was used to detect expressed FLAG-tagged ARL15 proteins. SDS-PAGE and Coomassie staining was employed to visualize and demonstrate equivalent amount of GST and GST-fused CBS-pair domain in the assay. SDS-PAGE and western blot was used to demonstrate equivalent amount of FLAG-tagged ARL15 WT and mutants as inputs in the pulldown assay. SDS-PAGE and western blot also used to analyze the binding of FLAG-tagged ARL15 WT and mutants to GST compared to GST-fused CBS-pair domain.

## Cell lines

Cultured human cells were maintained in a Dulbecco's Modified Eagle Medium (DMEM), high glucose media with 10% fetal bovine serum in a humidified 37°C, 5% $CO_2$ incubator. We used the LTRPC7 cell line (*Nadler et al., 2001*), herein referred to as 293-TRPM7 cells, expressing murine FLAG-tagged TRPM7 in response to tetracycline. The cells were generously provided by Dr. Andrew Scharenberg (University of Washington) and have been authenticated by SDS-PAGE and Western blotting. This cell line tested negative for mycoplasma contamination. The 293T cell line, here referred as HEK-293T cells, was purchased from ATCC (CRL-3216, Manassas, Virginia, USA). The authentication has been done by the vendor and this cell line also tested negative for mycoplasma contamination. The HEK293T cells used in CNNM $Mg^{2+}$ efflux assays were obtained from the laboratory of Nahum Sonenberg (McGill University) and tested negative for mycoplasma contamination.

## Co-immunoprecipitation

A 10 cm dish of HEK-293T cells were transfected with 8 µg of FLAG-tagged human ARL15 WT or the FLAG-tagged ARL15 R95A mutant (R95A) using the Turbofect Transfection Reagent. All the following biochemical procedures were conducted at 4°C. 24 hr post-transfection cells were lysed with 800 µl of mild lysis buffer (50 mM TRIS pH 7.4, 150 mM NaCl, 1% Igepal 630) containing protease inhibitors. Proteins were solubilized by incubating the lysis mixture for 30 min. The samples were then cleared by centrifugation at 15,600×$g$ for 10 min. Supernatants (lysates) were subjected to immunoprecipitation as follows. FLAG-tagged ARL15 proteins were immunoprecipitated using 50 µl of Pierce Anti-DYKDDDDK Magnetic Agarose (Thermo Fisher Scientific) for 2 hr at 4°C. The beads were washed two times with PBS and once with purified water. The bound proteins were eluted with 50 µl of 2× Laemmli sample buffer, and the proteins resolved by SDS-PAGE and western blotting using standard procedures. The anti-FLAG M2 antibody (Sigma-Aldrich) was used to detect FLAG-tagged ARL15 proteins. The anti-CNNM3 antibody (NBP2-32134, Novus Biologicals) was used to detect endogenous human CNNM3.

## Magnesium Green $Mg^{2+}$ efflux assay

HEK-293T cells were grown in a 3.5 cm glass bottom dish with a poly-lysine coating to ensure adherence. Cells were transfected with mCNNM2-mCherry fusion proteins and FLAG-tagged ARL15 (both WT and mutant) at a ratio of 1:3, mCNNM2-mCherry:ARL15. To ensure good cell separation, transfection was performed at densities between 20% and 30% using Lipofectamine 3000. Transfected cells were grown for 24 hr in DMEM media supplemented with 40 mM $MgCl_2$. Cells were then incubated in $Mg^{2+}$ loading buffer (78.1 mM NaCl, 5.4 mM KCl, 1.8 mM $CaCl_2$, 40 mM $MgCl_2$, 5.5 mM glucose, 5.5 mM HEPES, pH to 7.4 with KOH) supplemented with 2 mM Magnesium Green dye (Fisher Scientific) and 0.025% Pluronic f-127 (Fisher Scientific) in DMSO for 30 min at 37°C. Following incubation, the cells were rinsed once with the $Mg^{2+}$ loading buffer at room temperature and viewed via microscopy (AxioObserver.Z1 with X-cite series 120Q Illuminator, AxioCamMR3 camera, under the control of Axiovision software). Magnesium Green fluorescence (excitation 450–488 nm and emission 500–548 nm) was measured for 5 min (Filter set ET-GFP 49002, objective ET Plan-Neofluar 10×/0.20 Ph 1) with 20 s between frames. At 1 min, the buffer was changed to $Mg^{2+}$-free buffer (138.1 mM NaCl, 5.4 mM KCl, 1.8 mM $CaCl_2$, 5.5 mM glucose, 5.5 mM HEPES, pH to 7.4 with KOH). Following the efflux assay, CNNM2-mCherry expression was detected by measuring fluorescence from the ET-mCherry

Texas Red 49008 filter set (excitation 540–580 nm and emission 590–670 nm). Cell were then fixed by 3.7% formaldehyde and FLAG-tagged ARL15 proteins detected with anti-FLAG antibody (1:2000, F7425, Sigma-Aldrich) and Alexa 488-conjugated goat anti-rabbit secondary antibodies (A-11034, Thermo Fisher Scientific). Fluorescence in 10 cells expressing the proteins of interest was quantified using Fiji analysis software and the average plotted as function of time.

## Confocal microscopy

HEK-293T cells were grown on a microscope coverslip with a poly-lysine coating. Cells were transfected and antibody tagged as in the Magnesium Green efflux assay. Slides were then mounted using Diamond Anti-fade mounting and allowed to dry for 48 hr. mCherry and Alexa 488 fluorescence was imaged at 63× magnification by laser scanning confocal microscopy at the McGill University Advanced BioImaging Facility.

## Electrophysiological recordings

The voltage-clamp technique was used to evaluate the whole-cell currents of TRPM7 expressed in 293-TRPM7 cells as previously described (*Li et al., 2007*). Briefly, whole-cell current recordings of TRPM7-expressing cells were elicited by voltage stimuli lasting 250 ms delivered every 1 s using voltage ramps from –100 to +100 mV. Data were digitized at 2 or 5 kHz and digitally filtered offline at 1 kHz. The internal pipette solution for macroscopic current recordings contained 145 mM Cs methanesulfonate, 8 mM NaCl, 10 mM EGTA, and 10 mM HEPES, pH adjusted to 7.2 with CsOH. The extracellular solution for whole-cell recordings contained 140 mM NaCl, 5 mM KCl, 2 mM $CaCl_2$, 10 mM HEPES, and 10 mM glucose, pH adjusted to 7.4 with NaOH.

## $Zn^{2+}$ influx assay

The $Zn^{2+}$ influx assay used to characterize TRPM7 channel function has been previously described in detail (*Bai et al., 2021*). Briefly, cells were plated into 24-well dishes coated with polylysine to aid comparison between individual samples by fluorescence microscopy. Before labeling, cells were washed with Hanks' balanced salt solution (HBSS) containing 0.137 M NaCl, 5.4 mM KCl, 0.25 mM $Na_2HPO_4$, 6 mM glucose, 0.44 mM $KH_2PO_4$, 1.3 mM $CaCl_2$, 1.0 mM $MgSO_4$, and 4.2 mM $NaHCO_3$. Cells were then labeled with the $Zn^{2+}$ indicator FluoZin-3 (2.5 µM) in HBSS following the manufacturer's instructions (Thermo Fisher Scientific). The cells were then washed once with HBSS and then placed back into HBSS before images were quickly acquired on an inverted Olympus IX70 fluorescence microscope with a 10v phase contrast objective (Olympus Neoplan 10/0.25 Ph, Olympus, Tokyo, Japan). Cells were visually inspected for uneven dye loading prior to imaging. To stimulate $Zn^{2+}$ influx, 30 µM $ZnCl_2$ in HBSS was introduced to cells. For static measurements, images of cells from the different samples were taken at a specific time point 5–10 min after the addition of 30 µM $ZnCl_2$.

## Acknowledgements

We thank Tara Sprules, QANUC NMR Facility, for acquisition and analysis of the phosphorus NMR spectra, Katalin Illes and Alexei Gorelik, McGill University Biochemistry Dept., for assistance with cell biology and crystallographic data collection and processing. X-ray data were acquired at the Advanced Photon Source, a U.S. Department of Energy, Office of Science user facility operated by Argonne National Laboratory under contract DE-AC02-06CH11357. Funding: The study has been supported by funding from Natural Sciences and Engineering Research Council of Canada grant RGPIN-2020-07195 to KG and by the National Institutes of Health grant R01HL147350 to LR and LY.

## Additional information

### Funding

| Funder | Grant reference number | Author |
|---|---|---|
| Natural Sciences and Engineering Research Council of Canada | RGPIN-2020-07195 | Kalle Gehring |
| National Institutes of Health | R01HL147350 | Lixia Yue<br>Loren Runnels |

The funders had no role in study design, data collection and interpretation, or the decision to submit the work for publication.

### Author contributions

Luba Mahbub, Formal analysis, Investigation, Visualization, Methodology, Writing – original draft; Guennadi Kozlov, Formal analysis, Supervision, Investigation, Visualization, Methodology, Writing – original draft; Pengyu Zong, Emma L Lee, Sandra Tetteh, Formal analysis, Investigation, Visualization, Methodology; Thushara Nethramangalath, Investigation, Methodology; Caroline Knorn, Jianning Jiang, Ashkan Shahsavan, Investigation; Lixia Yue, Resources, Supervision, Visualization, Methodology; Loren Runnels, Conceptualization, Resources, Supervision, Visualization, Methodology; Kalle Gehring, Conceptualization, Supervision, Funding acquisition, Visualization, Project administration, Writing - review and editing

### Author ORCIDs

Luba Mahbub ⓘ http://orcid.org/0000-0001-7604-6134
Guennadi Kozlov ⓘ http://orcid.org/0000-0002-7742-6558
Caroline Knorn ⓘ http://orcid.org/0000-0001-8275-839X
Loren Runnels ⓘ http://orcid.org/0000-0002-2537-7360
Kalle Gehring ⓘ http://orcid.org/0000-0001-6500-1184

### Decision letter and Author response

Decision letter https://doi.org/10.7554/eLife.86129.sa1
Author response https://doi.org/10.7554/eLife.86129.sa2

## Additional files

### Supplementary files

• MDAR checklist

### Data availability

Diffraction data have been deposited in PDB under the accession code 8F6D.

The following dataset was generated:

| Author(s) | Year | Dataset title | Dataset URL | Database and Identifier |
|---|---|---|---|---|
| Kozlov G, Gehring K | 2022 | Crystal structure of the CNNM2 CBS-pair domain in complex with ARL15 | https://www.rcsb.org/structure/8f6d | RCSB Protein Data Bank, 8F6D |

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
