## [Editor Report]

In this important work, Mahbub and colleagues examine how the small GTPase ARL15 regulates ion flux through the proposed CNNM-TRPM7 complex. Using a complementary array of techniques, the authors gathered solid evidence for the binding of ARL15 to CNNM, resulting in a proposal on how this may affect the function of the CNNM-TRPM7 complex.

---

## [Decision Letter]

**Decision letter after peer review:**

Thank you for submitting your article "Structural insights into regulation of TRPM7 divalent cation uptake by the small GTPase ARL15" for consideration by *eLife*. Your article has been reviewed by 3 peer reviewers, and the evaluation has been overseen by a Reviewing Editor and Richard Aldrich as the Senior Editor. The reviewers have opted to remain anonymous.

Essential revisions:

The reviewers were generally enthusiastic about the potential of the work, but felt that a few additional experiments (and a more nuanced discussion) were required:

1) The authors obtained the CNNM-ARL15 complex structure, not the TRPM7-ARL15 structure, so it is important to test the effect of ARL15 on Mg^2+^ export by CNNM. This could be achieved either via Mg-sensitive probes or transport assays.

2) Similarly, the functional studies do not demonstrate whether ARL15 regulates TRPM7 function via CNNM and/or independently of CNNM proteins. This should be addressed through for example mutant proteins/KO cells, and/or selective inhibitors.

3) The authors state that ARL15 could not be loaded with guanine nucleotide and does not have intrinsic GTPase activity. This lack of GTPase activity is puzzling, as others have recently shown that Arl15 can bind to guanine nucleotide and hydrolyze GTP in vitro (Shi et al., *eLife*, 2022). Therefore, this issue requires further attention and discussion. At the very least, the authors should detail their protocols and elaborate on the discrepancy.

4) The absence of both guanine nucleotide and magnesium in the solved crystal structure of Arl15 is a potential concern. Guanine nucleotide and magnesium binding can significantly change the structures of the switch I, II, and inter-switch toggle of a small GTPase. Therefore, nucleotide-free small GTPases might not be physiological and could be problematic for interpreting the co-structure. This needs to be addressed.

5) Lastly, the role of (Mg)ATP binding to the CBS domains in CNNM needs to be clarified (see recommendations to authors by reviewer #2 for details, please also add statistics).

*Reviewer #1 (Recommendations for the authors):*

In this paper, the authors determined the crystal structure of ARL15 in complex with CNNM2 CBS domain and performed the mutational analysis based on the structure, which revealed the binding mechanism between ARL15 and CNNM. In addition, the authors showed that the R95A mutant of ARL15 did not inhibit the function of TRPM7.

However, the detailed mechanism of TRPM7 inhibition by ARL15 remains unclear because the structure of TRPM7 in complex with ARL15 is still unknown. Furthermore, despite the structure determination of ARL15 in complex with CNNM, the effect of ARL15 on CNNM function is still unclear.

Therefore, given the limited impact of the work, I would recommend this for publication if the authors can further strengthen the impact of the work, for example by characterizing the effect of ARL15 on CNNM transporters, which is feasible for the authors. Please also address the following concerns.

1. Title, "Structural insights into regulation of TRPM7 divalent cation uptake by the small GTPase ARL15".

The title makes it sound as if the authors have determined the structure of the TRPM7-ARL15 complex, which is not the case. Please change the title to one that includes the word "CNNM".

2. Page 4, lines 3-6. "Structurally, CNNMs consist of an N-terminal extracellular domain, a transmembrane domain, and two cytosolic domains: a CBS pair domain (also termed a Bateman domain) and a cyclic nucleotide-binding homology (CNBH) domain (de Baaij et al., 2012)."

In addition to citing the CNNM structure papers, please cite the literature on the cryo-EM structure of TRPM7 (10.1073/pnas.1810719115) in this paragraph and state that the structure of the TRPM7-ARL15 complex has not yet been reported.

3. Page 4, lines 6-8. "Two structures of orthologous prokaryotic (CorB) proteins have been determined, confirming that the CNNMs are ion transporters (Chen et al., 2021; Huang et al., 2021)."

Of the two papers cited here, one is called CorB, while the other is called CorC. So please change the phrase from "Two structures of orthologous prokaryotic (CorB) proteins" to "Two structures of orthologous prokaryotic (CorB/CorC) proteins" or just "Two structures of orthologous prokaryotic proteins". Either is fine.

4. Page 6, lines 14-19. "We also tested if Mg^2+^-ATP affects ARL15 binding. Mg^2+^-ATP binds to CNNM CBS-pair domains and promotes dimerization in a flat ring conformation (Corral-Rodriguez et al., 2014). We observed no change in ARL15 affinity when Mg^2+^-ATP was present in agreement with co-immunoprecipitation experiments (Zolotarov et al., 2021). This suggests that CBS-pair dimerization does not regulate ARL15-binding".

According to your ITC data (Figure 2D and Figure S1.), the affinity is similar or slightly higher in the presence of ATP. However, in the following section (page 8, lines 4-8), the authors also described it as follows, which rather implies that the binding affinity might be lower in the presence of ATP because the authors stated that the ATP-dependent structural changes would generate the contact surfaces.

"In the crystal, the ARL15 makes contact with two CBS-pair domains (Figure 2E). The CBS-pair domains are present as a dimer but compared to previous structures, the dimer is twisted, possibly due to the absence of Mg^2+^-ATP. This twist opens the dimer slightly generating two contact surfaces between ARL15 and the CNNM2 CBSpair domains."

Therefore, I find this structural change interesting, but I am also a little confused by the structural interpretation of the authors. Please revise these parts (page 6, lines 14-19 )(page 8, lines 4-8) to avoid confusion. In addition, please make a new figure to show the ARL15-dependent structural changes of CNNM and how the contacting surfaces form.

5. Page 12. CNNM-binding-defective ARL15 is unable to inhibit TRPM7 channel activity.

In Figure 6, the authors showed that overexpression of the ARL15 R95A mutant failed to suppress TRPM7-dependent currents, whereas overexpression of ARL15 WT suppressed them, and hypothesized that this inhibition was mediated by CNNM binding.

However, in this experiment, while both ARL15 and TRPM7 were overexpressed, CNNM was not. Thus, I am not sure whether the endogenous expression level of CNNM is sufficient to suppress overexpressed TRPM7. Alternatively, it might be possible that the R95A mutation directly affects the binding of ARL15 to TRPM7. Therefore, I would strongly encourage the authors to test the co-IP of TRPM7 with ARL15 and ARL15 R95A in the revised manuscript.

6. Page 16, lines 9-13. "The model suggests that ARL15-binding may stimulate CNNM activity. This is untested but reasonable given that ARL15 can displace PRLs from CNNMs. As CNNMs are electroneutral ion antiporters (Chen et al., 2021; Yamazaki et al., 2013), detection of their activity in electrophysiological experiments is more difficult than detecting TRPM7 activity."

In the previous publication by the authors themselves (10.1016/j.str.2019.11.016), they performed fluorescence-based Mg^2+^ efflux assays of CNNM. Thus, the evaluation of the effect of ARL15 on CNNM-dependent Mg^2+^ efflux should be feasible for the authors, and it would be more appropriate to evaluate the structure of the ARL15-CNNM complex. Therefore, I find it quite strange that the authors did not perform it, and also suggest the authors perform it to strengthen the impact of this work for publication in *eLife*.

7. Page 19, line 17. "(Supp. Table X)"

This should be Table 1.

8. Page 19, line 21. "(Senior et al., 2020)"

Please cite the AlphaFold2 paper (10.1038/s41586-021-03819-2).

9. Reference format. Page 12, line 10 ({Gimenez-Mascarell, 2017 #216}).　Page 16, line 7 ({Bai, 2021 #1713}).

These are unformatted. Please format them.

10. Full wwPDB X-ray Structure Validation Report (Full wwPDB X-ray Structure Validation Report) and Supporting Zip File "maps and coordinates for review"

The Ramachandran outlier ratio in the validation report is relatively high (0.5%) and does not match the out-of-bounds regions in the table (0.4%). I would recommend the authors correct the models to reduce the outlier.

*Reviewer #2 (Recommendations for the authors):*

1. Previous work from the group of dr. Martinez-Cruz has demonstrated that the CBS domains of CNNM2 can be in a "flat" or "twisted" confirmation. The flat confirmation is obtained by ATP binding and can be locked by binding of PRLs. The patient-derived T568I mutation also locks the CBS domain in their "flat" confirmation. Can ARL15 bind to the CBS domains in both the "flat" and the "twisted" confirmation? Do the current crystallization conditions (without ATP) result in a preferential "twisted" confirmation of the CBS domains? And how does this affect the ARL15 binding?

2. The authors describe that the addition of GTP and GDP does not improve CNNMcbs-ARL15. Interestingly, ATP does seem to "improve" affinity between ARL-CNNM2, factor two difference, although the authors mention in the result section it does not (statistics have not been described in the figure legend). This is interesting, as ATP would result in a conformational change towards the "flat" state. Is the ATP effect significantly different? How would to authors interpret this result?

3. Could to authors perform the competitive (PRL2/ARL15) binding assays using the CNNM2-T568I mutant? As this mutant locks the cbs domains in the flat state (which would be the predominant state in ATP conditions anyway), it would be important to interpret the relevance of the binding assays.

4. Multiple studies have demonstrated that ARL15 predominantly residues in the Golgi (Rocha et al. 2017, Wu et al. 2021, Zolotarov et al.). Actually, palmitoylation of ARL15 is required for proper localization to the Golgi apparatus and this sequence was deleted in your constructs. How does this interfere with your interpretations? CNNM and TRPM7 proteins are likely to go through the Golgi as well but are primarily plasma membrane proteins. Where does ARL15 interact with CNNM? And where does PRL interact with CNNM? Could the authors include localization studies for ARL15 (wild type and S94K), TRPM7, and CNNM? This might help to improve the model you describe in figure 7.

5. The functional studies do not demonstrate whether ARL15 regulates TRPM7 function via CNNM and/or independently of CNNM proteins. Interaction studies, GST or Co-IP, would be required to see if TRPM7 still binds to CNNMs proteins when ARL15 is mutated. Bai et al. have shown with various CNNM KO cell models that TRPM7 is dependent on their presence. I would recommend doing the Zn2+ transport assays in the absence of CNNM proteins. If indeed ARl15 only acts via CNNM proteins, CNNM KO cells should show similar results to WT-ARL15 and R95W-ARl15 overexpression in patch-clamp or FluoZin uptake assays.

6. TRPM7 does not only transport Zn2+, but also Mg^2+^ (Zn2+ has a higher affinity, but TRPM7 mainly transports Mg^2+^ in vitro/vivo). To support your model in figure 7, it is vital to include other ion transport assays as well. What is the intracellular Mg^2+^ concentration in these cells? Please include Magfura-2/MagnesiumGreen and or 25Magnesium transport assays (Bai et al). Does ARL15 expression change upon different intracellular Mg^2+^ concentrations?

7. What happens to PRL expression/binding to CNNM proteins in cells transfected with ARL15 mutants? Is this increased upon loss of functional ARL15? Does this correspond with the intracellular Mg^2+^ concentration?

*Reviewer #3 (Recommendations for the authors):*

Although the interaction between Arl15, CNNMs, and TRPM7 has been reported, it is still unclear how they interact at the molecular level. The authors systematically and quantitatively studied Arl15-CNNM interaction using biochemical and biophysical approaches. First, they found that Arl15 is an unusual Arl GTPase because it has a very low affinity toward GTP and shows no GTPase activity. Next, they demonstrated the interaction between Arl15 and the CBS-pair domain using the ITC and found that the interaction is independent of the guanine nucleotide-binding status of Arl15. The author then resolved the crystal structure of Arl15 and the CNNM2 CBS-pair domain complex with a resolution of 3.2 angstroms. Their result differed from the computed structure Zolotarov et al. reported (Cell. Mol. Life Sci. 2021). Next, guided by the atomic details of the interaction interface, the author made point mutations and verified their structure by extensive in vitro pulldowns. Since PRLs also interact with the CBS-pair domain, the authors found that the binding of the CBS-pair domain to PRLs and Arl15 is mutually exclusive. At last, using the point mutations of CNNM, they showed that Arl15-CNNM binding is required for Arl15 regulation of TRPM7 in zinc ion influx assay.

Comments and concerns

1) The authors' claim on the lack of both guanine nucleotide binding and GTPase activities in Arl15 is unconvincing. Interestingly, Shi et al. (*eLife*, 2022) have reported the GTP-binding and GTP-hydrolysis of Arl15 using purified proteins. The authors also should carefully address the below issues.

Regarding the guanine nucleotide-binding of Arl15, the authors stated that "Experiments to preload ARL15 with GTP or GDP were unsuccessful, and we observed that the protein purified from *E. coli* was nucleotide-free." It is unclear what experimental procedure the authors employed to load Arl15 with guanine nucleotides. Also, it is unknown how they determined that purified Arl15 was nucleotide-free. The authors measured the affinity of Arl15 toward GTP as 20 uM, which seems sufficient to maintain an almost fully loaded Arl15-GTP in cells, given that the GTP concentration in cells is ~ 0.2 mM. Hence, it is puzzling why Arl15 purified from cells are nucleotide-free and cannot be loaded with guanine nucleotide. The author stated, "Attempts to detect GTPase activity were unsuccessful either due to the absence of a GTPase activating protein (GAP) or an intrinsic lack of catalytic activity." The manuscript did not disclose how the authors performed the GTP hydrolysis assay and what positive and negative controls were used. Note that failed GTPase assay should not be evidence to demonstrate the lack of GTPase activity of Arl15.

2) My primary concern is the lack of description of the nucleotide and magnesium in Arl15 in the resolved crystal structure. What is the guanine nucleotide-binding status of Arl15 in the crystal structure? Is low binding affinity between Arl15 and CBS due to the lack of GTP? Since the authors claimed that Arl15 binds GTP with the Kd of 20 uM, Arl15 should be mostly GTP-bound in vivo since the cellular concentration of GTP is ~ 0.2 mM, which is 10-fold that of GDP. Therefore, the structure of nucleotide-free Arl15 complexed with the CBS-pair domain probably does not reflect their in vivo binding.

3) Figure 2. The schematic domain organization diagram of CNNMs is too simplified. There are three transmembrane domains. Most importantly and highly relevant to this study, the CBS-pair domain consists of CBS1 + CBS2. Figure 2E should present multiple rotational views of the Arl15-CBS complex with clear labeling of CBS1, CBS2, and Arl15's switch I and II regions. In Figure 2, it would benefit the readers if the authors also present primary sequences of the CBS-pair domain and Arl15 with key amino acids annotated.

4) The authors might comment on the following questions in the Discussion. How does Arl15 regulate the activity of TRPM7? Is it via a direct interaction between Arl15 and TRPM7? Or is it via CNNMs? How do CNNMs transduce the signal from Arl15 to TRPM7?

---

## [Author Response]

Essential revisions:The reviewers were generally enthusiastic about the potential of the work, but felt that a few additional experiments (and a more nuanced discussion) were required:

We thank the reviewers for their insightful and helpful comments and are pleased to submit a revised manuscript for their consideration. We have included additional experiments to characterize nucleotide binding and the effect of ARL15 on CNNM magnesium transport.

1) The authors obtained the CNNM-ARL15 complex structure, not the TRPM7-ARL15 structure, so it is important to test the effect of ARL15 on Mg^2+^ export by CNNM. This could be achieved either via Mg-sensitive probes or transport assays.

As requested by multiple reviewers, we have performed Mg^2+^ efflux assays to test the effect of ARL15. We observed that wild-type ARL15 inhibits CNNM2 efflux while the ARL15 R95A mutant does not. Thus, ARL15 appears to act as a general inhibitor of CNNM/TRPM7 transport while PRLs are selective, inhibiting CNNM efflux but promoting TRPM7 influx.

2) Similarly, the functional studies do not demonstrate whether ARL15 regulates TRPM7 function via CNNM and/or independently of CNNM proteins. This should be addressed through for example mutant proteins/KO cells, and/or selective inhibitors.

At present, we cannot distinguish between direct and indirect inhibition. We only know that the CNNM-ARL15 interface blocks the stimulation of TRPM7 by CNNM. As suggested by Reviewer 1, we have modified the title to better reflect that fact. Without a structure of the complex with TRPM7, the mechanism of inhibition is difficult to address. It seems likely that ARL15 makes contacts with TRPM7 but whether these are strong enough to detect in co-IP or not doesn't provide insight into the mechanism of regulation.

3) The authors state that ARL15 could not be loaded with guanine nucleotide and does not have intrinsic GTPase activity. This lack of GTPase activity is puzzling, as others have recently shown that Arl15 can bind to guanine nucleotide and hydrolyze GTP in vitro (Shi et al., eLife, 2022). Therefore, this issue requires further attention and discussion. At the very least, the authors should detail their protocols and elaborate on the discrepancy.

It is impossible to prove that ARL15 never has GTPase activity given that we can't exclude a GAP exists. However, the evidence clearly points toward ARL15 acting simply as a GTP-binding protein. In the revised manuscript, we have included data that shows that (1) purified recombinant ARL15 is nucleotide-free, (2) the binding of GTP, and (3) GDP does not accumulate when ARL15 is incubated with GTP. We were aware of the report by Shi et al. but have been unable to reproduce their results. We now cite the paper and have added a description of the discrepancies between the two reports.

4) The absence of both guanine nucleotide and magnesium in the solved crystal structure of Arl15 is a potential concern. Guanine nucleotide and magnesium binding can significantly change the structures of the switch I, II, and inter-switch toggle of a small GTPase. Therefore, nucleotide-free small GTPases might not be physiological and could be problematic for interpreting the co-structure. This needs to be addressed.

We agree this is an important issue. ARL15 does undergo a conformational change upon GTP binding (Figure 1C) but this does not affect CNNM binding of the purified proteins. Measurements with or without GTP or magnesium reproducibly show no difference in affinity. Intriguingly, we did observe a small 40% increase in the K_d_ in the presence of GDP (see Figure 2d); however, the significance of this is unclear given the cellular concentration of GTP is ~10-fold greater than GDP. Physiologically, the vast majority of ARL15 will have GTP bound.

On the other hand, the difference between the GTP-bound and nucleotide-free protein did appear to affect crystallization. We observed better crystal growth in the presence of a non-hydrolysable GTP analog, although the analog was not present in the crystals. The most probable explanation is that the contacts responsible for crystallization (crystal packing forces) prevent nucleotide binding. The improved quality of the crystals in the presence of the nucleotide could be due to slower crystallization or improved protein stability.

5) Lastly, the role of (Mg)ATP binding to the CBS domains in CNNM needs to be clarified (see recommendations to authors by reviewer #2 for details, please also add statistics).

We have repeated the affinity measurements and confirm that ATP enhances the binding affinity two-fold. As suggested by the reviewer, we used a CBS-pair domain mutant (T568I) that is unable to bind ATP and can now attribute the affinity different to ATP binding site to the CNNM2 CBS-pair domain.

Reviewer #1 (Recommendations for the authors):In this paper, the authors determined the crystal structure of ARL15 in complex with CNNM2 CBS domain and performed the mutational analysis based on the structure, which revealed the binding mechanism between ARL15 and CNNM. In addition, the authors showed that the R95A mutant of ARL15 did not inhibit the function of TRPM7.However, the detailed mechanism of TRPM7 inhibition by ARL15 remains unclear because the structure of TRPM7 in complex with ARL15 is still unknown. Furthermore, despite the structure determination of ARL15 in complex with CNNM, the effect of ARL15 on CNNM function is still unclear.Therefore, given the limited impact of the work, I would recommend this for publication if the authors can further strengthen the impact of the work, for example by characterizing the effect of ARL15 on CNNM transporters, which is feasible for the authors. Please also address the following concerns.

We thank the reviewer for the useful suggestion. We now report the effect of ARL15 on CNNM2 Mg^2+^ transport activity (detailed below).

1. Title, "Structural insights into regulation of TRPM7 divalent cation uptake by the small GTPase ARL15".The title makes it sound as if the authors have determined the structure of the TRPM7-ARL15 complex, which is not the case. Please change the title to one that includes the word "CNNM".

We thank the reviewer for the suggestion and have added CNNM to the title as "Structural insights into regulation of TRPM7/CNNM divalent cation uptake by the small GTPase ARL15". This better reflects the structure and functional assays in the manuscript.

2. Page 4, lines 3-6. "Structurally, CNNMs consist of an N-terminal extracellular domain, a transmembrane domain, and two cytosolic domains: a CBS pair domain (also termed a Bateman domain) and a cyclic nucleotide-binding homology (CNBH) domain (de Baaij et al., 2012)."In addition to citing the CNNM structure papers, please cite the literature on the cryo-EM structure of TRPM7 (10.1073/pnas.1810719115) in this paragraph and state that the structure of the TRPM7-ARL15 complex has not yet been reported.

We have added a citation to the TRPM7 structure. We additionally cite two new publications that have appeared since our initial submission: a second TRPM7 structure (Nadezhdin et al., Nat Comm, 2023) and a study of the effect of PRL phosphatase on TRPM7 activity (Hardy et al., PNAS, 2023).

3. Page 4, lines 6-8. "Two structures of orthologous prokaryotic (CorB) proteins have been determined, confirming that the CNNMs are ion transporters (Chen et al., 2021; Huang et al., 2021)."Of the two papers cited here, one is called CorB, while the other is called CorC. So please change the phrase from "Two structures of orthologous prokaryotic (CorB) proteins" to "Two structures of orthologous prokaryotic (CorB/CorC) proteins" or just "Two structures of orthologous prokaryotic proteins". Either is fine.

We have changed the text.

4. Page 6, lines 14-19. "We also tested if Mg^2+^-ATP affects ARL15 binding. Mg^2+^-ATP binds to CNNM CBS-pair domains and promotes dimerization in a flat ring conformation (Corral-Rodriguez et al., 2014). We observed no change in ARL15 affinity when Mg^2+^-ATP was present in agreement with co-immunoprecipitation experiments (Zolotarov et al., 2021). This suggests that CBS-pair dimerization does not regulate ARL15-binding".According to your ITC data (Figure 2D and Figure S1.), the affinity is similar or slightly higher in the presence of ATP. However, in the following section (page 8, lines 4-8), the authors also described it as follows, which rather implies that the binding affinity might be lower in the presence of ATP because the authors stated that the ATP-dependent structural changes would generate the contact surfaces."In the crystal, the ARL15 makes contact with two CBS-pair domains (Figure 2E). The CBS-pair domains are present as a dimer but compared to previous structures, the dimer is twisted, possibly due to the absence of Mg^2+^-ATP. This twist opens the dimer slightly generating two contact surfaces between ARL15 and the CNNM2 CBSpair domains."Therefore, I find this structural change interesting, but I am also a little confused by the structural interpretation of the authors. Please revise these parts (page 6, lines 14-19) (page 8, lines 4-8) to avoid confusion. In addition, please make a new figure to show the ARL15-dependent structural changes of CNNM and how the contacting surfaces form.

We apologize for the lack of clarity. We have repeated the experiments and can confirm that there is a small, two-fold improvement in ARL15-CNNM binding in the presence of ATP. Furthermore, as suggested by Reviewer #2, we carried out additional experiments with a CNNM2 mutant that is unable to bind ATP. The mutation abrogated the affinity change, thus showing the two-fold improvement is due to ATP binding to the CNNM2 CBS-pair domain.

We hesitate to draw firm conclusions about the physiological relevance for several reasons. (1) The affinity difference is relatively small and CBS-pair domain still binds ARL15 with low μM affinity even in the absence of ATP. It would be an exaggeration to claim at this point that CBS-pair dimerization *regulates* ARL15-binding. (2) The affinity measurements were made using cytosolic fragments (either the CBS-pair domain alone or with the CNBH domain present). The affinities between the intact proteins on the membrane could easily change by more than two-fold. (3) The ATP concentration in the cell cytosol is high enough that ATP is expected to be always bound to the CBS-pair domain.

As requested, we have added a supplemental figure S3 to show the range of conformations of CNNM2 dimers.

5. Page 12. CNNM-binding-defective ARL15 is unable to inhibit TRPM7 channel activity.In Figure 6, the authors showed that overexpression of the ARL15 R95A mutant failed to suppress TRPM7-dependent currents, whereas overexpression of ARL15 WT suppressed them, and hypothesized that this inhibition was mediated by CNNM binding.However, in this experiment, while both ARL15 and TRPM7 were overexpressed, CNNM was not. Thus, I am not sure whether the endogenous expression level of CNNM is sufficient to suppress overexpressed TRPM7. Alternatively, it might be possible that the R95A mutation directly affects the binding of ARL15 to TRPM7. Therefore, I would strongly encourage the authors to test the co-IP of TRPM7 with ARL15 and ARL15 R95A in the revised manuscript.

Unfortunately, due to technical difficulties, we were unable to test the co-IP. We agree this would be a useful experiment for future follow up.

6. Page 16, lines 9-13. "The model suggests that ARL15-binding may stimulate CNNM activity. This is untested but reasonable given that ARL15 can displace PRLs from CNNMs. As CNNMs are electroneutral ion antiporters (Chen et al., 2021; Yamazaki et al., 2013), detection of their activity in electrophysiological experiments is more difficult than detecting TRPM7 activity."In the previous publication by the authors themselves (10.1016/j.str.2019.11.016), they performed fluorescence-based Mg^2+^ efflux assays of CNNM. Thus, the evaluation of the effect of ARL15 on CNNM-dependent Mg^2+^ efflux should be feasible for the authors, and it would be more appropriate to evaluate the structure of the ARL15-CNNM complex. Therefore, I find it quite strange that the authors did not perform it, and also suggest the authors perform it to strengthen the impact of this work for publication in eLife.

We thank the reviewer for the suggestion. The experiment was planned, and we had indeed expected that ARL15 might stimulate CNNM activity. Instead, we find that ARL15 robustly inhibits CNNM2 Mg^2+^ efflux. The inhibition depends on ARL15 binding to CNNM2 as the R95A mutant shows no inhibition. The results have been added as Figure 7A and the model in Figure 8 updated. Experiments confirming the localization of CNNM2 and ARL15 are presented in the supplements to Figure 7.

7. Page 19, line 17. "(Supp. Table X)"This should be Table 1.

Corrected.

8. Page 19, line 21. "(Senior et al., 2020)"Please cite the AlphaFold2 paper (10.1038/s41586-021-03819-2).

Corrected.

9. Reference format. Page 12, line 10 ({Gimenez-Mascarell, 2017 #216}).　Page 16, line 7 ({Bai, 2021 #1713}).These are unformatted. Please format them.

Corrected.

10. Full wwPDB X-ray Structure Validation Report (Full wwPDB X-ray Structure Validation Report) and Supporting Zip File "maps and coordinates for review"The Ramachandran outlier ratio in the validation report is relatively high (0.5%) and does not match the out-of-bounds regions in the table (0.4%). I would recommend the authors correct the models to reduce the outlier.

The electron density is not well defined in some parts of the structure. We've tried to eliminate as many of the outliers as possible, but they reappear post-refinement. According to the validation report, the fraction of outliers is slightly better than half of the structures of similar resolution.

**Author response image 1. sa2fig1:** 

The difference between the value in the Structure Validation Report and refinement table was due to rounding down. There are 5 outliers out of 1088 dihedrals = 4.59%. We've increased the number of decimal places in Table 1 to avoid the issue.

Reviewer #2 (Recommendations for the authors):1. Previous work from the group of dr. Martinez-Cruz has demonstrated that the CBS domains of CNNM2 can be in a "flat" or "twisted" confirmation. The flat confirmation is obtained by ATP binding and can be locked by binding of PRLs. The patient-derived T568I mutation also locks the CBS domain in their "flat" confirmation. Can ARL15 bind to the CBS domains in both the "flat" and the "twisted" confirmation? Do the current crystallization conditions (without ATP) result in a preferential "twisted" confirmation of the CBS domains? And how does this affect the ARL15 binding?

The reviewer raises an important question. Dimerization of CNNM CBS-pair domains is a complicated phenomenon and undoubtedly important for the regulation of ion transport. While we can't force the CBS-pair domains into a flat or twisted conformation, we can confirm that ATP improves ARL15 binding to CNNM2 CBS-pair domain.

We have repeated the original experiments and additionally used the CBS-pair domain T568I mutant characterized by the Martínez-Cruz group (Corral-Rodríguez, Biochem J, 2014) to show that the increase in affinity is directly a consequence of ATP binding (Figure 2 and associated supplemental material).

2. The authors describe that the addition of GTP and GDP does not improve CNNMcbs-ARL15. Interestingly, ATP does seem to "improve" affinity between ARL-CNNM2, factor two difference, although the authors mention in the result section it does not (statistics have not been described in the figure legend). This is interesting, as ATP would result in a conformational change towards the "flat" state. Is the ATP effect significantly different? How would to authors interpret this result?

We have repeated the experiments with GTP and GDP to confirm the reliability of the measurements. The repeated experiments are shown in the Supplemental Figure S2. To account for uncertainty in protein extinction coefficients and other sources of error, we use ±15% as an error estimate when reporting affinities.

3. Could to authors perform the competitive (PRL2/ARL15) binding assays using the CNNM2-T568I mutant? As this mutant locks the cbs domains in the flat state (which would be the predominant state in ATP conditions anyway), it would be important to interpret the relevance of the binding assays.

We have performed the competition experiment requested and observed that PRL2 inhibits ARL15 binding to the CNNM2 CBS-pair T568I mutant (Figure 6 supplemental figure).

4. Multiple studies have demonstrated that ARL15 predominantly residues in the Golgi (Rocha et al. 2017, Wu et al. 2021, Zolotarov et al.). Actually, palmitoylation of ARL15 is required for proper localization to the Golgi apparatus and this sequence was deleted in your constructs. How does this interfere with your interpretations? CNNM and TRPM7 proteins are likely to go through the Golgi as well but are primarily plasma membrane proteins. Where does ARL15 interact with CNNM? And where does PRL interact with CNNM? Could the authors include localization studies for ARL15 (wild type and S94K), TRPM7, and CNNM? This might help to improve the model you describe in figure 7.

These are important questions but largely out of the scope of the current study.

Based on the crystal structure and mutagenesis results, palmitoylation appears unlikely to affect the interaction between ARL15 and CNNMs. Pulldown and co-IP experiments in HEK-293T cells confirmed that the full-length protein binds to the CNNM2 GST-CBS-pair domain and can co-IP endogenous CNNM3 (Figure 6F and G). However, levels of ARL15 palmitoylation were not measured.

In agreement with previous reports, we observed CNNM2 and ARL15 on internal and plasma membranes in cells used to study CNNM2 Mg^2+^ efflux (Figure 7—supplement 1). The R95A mutant did not appear to affect ARL15 localization but it did prevent inhibition of CNNM-associated Mg^2+^ efflux.

It is not known at which point TRPM7, ARL15, PRLs and CNNM proteins interact following their synthesis. ARL15 was shown to affect CNNM3 glycosylation (Zolotarov et al., Cell Mol Life Sci, 2021) but the mechanism proposed in which ARL15 is within the ER lumen seems unlikely (Figure 7 of that work). We hope our study will prompt future groups to examine ARL15, TRPM7, and CNNM localization more closely.

5. The functional studies do not demonstrate whether ARL15 regulates TRPM7 function via CNNM and/or independently of CNNM proteins. Interaction studies, GST or Co-IP, would be required to see if TRPM7 still binds to CNNMs proteins when ARL15 is mutated. Bai et al. have shown with various CNNM KO cell models that TRPM7 is dependent on their presence. I would recommend doing the Zn2+ transport assays in the absence of CNNM proteins. If indeed ARl15 only acts via CNNM proteins, CNNM KO cells should show similar results to WT-ARL15 and R95W-ARl15 overexpression in patch-clamp or FluoZin uptake assays.

Again, these are excellent questions, but we can provide only partial answers. We know ARL15 binding to CNNM protein affects TRPM7 function but not how this occurs. As requested, we have carried out Mg^2+^ transport assays and show that ARL15 also inhibits CNNM efflux. Thus, ARL15 appears to function as a general inhibitor of CNNM/TRPM7 cation transport while PRLs inhibit CNNM efflux and stimulate TRPM7 influx. Future studies are planned to study TRPM7 function in CNNM KO cells.

6. TRPM7 does not only transport Zn2+, but also Mg^2+^ (Zn2+ has a higher affinity, but TRPM7 mainly transports Mg^2+^ in vitro/vivo). To support your model in figure 7, it is vital to include other ion transport assays as well. What is the intracellular Mg^2+^ concentration in these cells? Please include Magfura-2/MagnesiumGreen and or 25Magnesium transport assays (Bai et al). Does ARL15 expression change upon different intracellular Mg^2+^ concentrations?

We have included the Magnesium Green transport assays requested and have modified the model (now in Figure 8) in light of those results. We now show that ARL15 inhibits both CNNM and TRPM7. A recent publication (see below) suggests that PRL2 overexpression impacts ARL15 expression and that ARL15 protein stability is reduced when not in complex with CNNM3.

7. What happens to PRL expression/binding to CNNM proteins in cells transfected with ARL15 mutants? Is this increased upon loss of functional ARL15? Does this correspond with the intracellular Mg^2+^ concentration?

A recent publication from the Tremblay group examining TRPM7 regulation by PRLs (Hardy et al., PNAS, 2023) addresses many of these questions. They observed that ARL15 levels decrease when PRL2 is highly expressed and that ARL15 expression strengthened the association between CNNM3 and TRPM7. In agreement with our observations, they observed that PRL2 inhibits the ARL15 interaction with CNNM3.

Reviewer #3 (Recommendations for the authors):Although the interaction between Arl15, CNNMs, and TRPM7 has been reported, it is still unclear how they interact at the molecular level. The authors systematically and quantitatively studied Arl15-CNNM interaction using biochemical and biophysical approaches. First, they found that Arl15 is an unusual Arl GTPase because it has a very low affinity toward GTP and shows no GTPase activity. Next, they demonstrated the interaction between Arl15 and the CBS-pair domain using the ITC and found that the interaction is independent of the guanine nucleotide-binding status of Arl15. The author then resolved the crystal structure of Arl15 and the CNNM2 CBS-pair domain complex with a resolution of 3.2 angstroms. Their result differed from the computed structure Zolotarov et al. reported (Cell. Mol. Life Sci. 2021). Next, guided by the atomic details of the interaction interface, the author made point mutations and verified their structure by extensive in vitro pulldowns. Since PRLs also interact with the CBS-pair domain, the authors found that the binding of the CBS-pair domain to PRLs and Arl15 is mutually exclusive. At last, using the point mutations of CNNM, they showed that Arl15-CNNM binding is required for Arl15 regulation of TRPM7 in zinc ion influx assay.Comments and concerns1) The authors' claim on the lack of both guanine nucleotide binding and GTPase activities in Arl15 is unconvincing. Interestingly, Shi et al. (eLife, 2022) have reported the GTP-binding and GTP-hydrolysis of Arl15 using purified proteins. The authors also should carefully address the below issues.

Despite our best efforts, we have been unable to reproduce the results of Shi et al. We now cite the paper and have added a description of the discrepancies between the two reports.

Regarding the guanine nucleotide-binding of Arl15, the authors stated that "Experiments to preload ARL15 with GTP or GDP were unsuccessful, and we observed that the protein purified from *E. coli* was nucleotide-free." It is unclear what experimental procedure the authors employed to load Arl15 with guanine nucleotides. Also, it is unknown how they determined that purified Arl15 was nucleotide-free. The authors measured the affinity of Arl15 toward GTP as 20 uM, which seems sufficient to maintain an almost fully loaded Arl15-GTP in cells, given that the GTP concentration in cells is ~ 0.2 mM. Hence, it is puzzling why Arl15 purified from cells are nucleotide-free and cannot be loaded with guanine nucleotide. The author stated, "Attempts to detect GTPase activity were unsuccessful either due to the absence of a GTPase activating protein (GAP) or an intrinsic lack of catalytic activity." The manuscript did not disclose how the authors performed the GTP hydrolysis assay and what positive and negative controls were used. Note that failed GTPase assay should not be evidence to demonstrate the lack of GTPase activity of Arl15.

We have expanded Figure 1 to include the experiments (UV & NMR spectroscopy) that show purified ARL15 does not contain bound nucleotide. The reviewer is correct that in cells, ARL15 is likely fully loaded with GTP; however, during purification the ligand is rapidly lost since the protein concentration is less than the binding affinity (for an explanation of the ligand retention effect, see Silhavy et al., PNAS, 1975).

Our measured GTP binding affinity is similar to values reported for related proteins. The Wittinghofer laboratory reported ARL2 has an affinity of 0.17 µM for mant-GTP (Hanzal-Bayer et al., JMB, 2005). The published affinity of ARL3 is 48 µM (Linari et al., FEBS Letters, 1999). More recently, the affinity of ARL13B for mant-Gpp(NH)p was reported to be 0.4 µM (Ivanova et al., JBC, 2017).

As the reviewer correctly points out, it is impossible to disprove the ARL15 has GTPase activity in the presence of a GTPase activating protein (GAP); however, we can measure its intrinsic activity. Incubation of ARL15 saturated with GTP did not show significant accumulation of GDP (Suppl. Figure S1). We did detect an overall decrease overnight in the GTP concentration, possibly due to the presence of contaminating bacterial hydrolases. From the concentration of ARL15 (1 mM), we can put an upper bound on the intrinsic turnover rate of 0.000005 s^-1^. This is an order of magnitude smaller than the intrinsic GTPase activity of hRas (Neal et al., JBC, 1988).

2) My primary concern is the lack of description of the nucleotide and magnesium in Arl15 in the resolved crystal structure. What is the guanine nucleotide-binding status of Arl15 in the crystal structure? Is low binding affinity between Arl15 and CBS due to the lack of GTP? Since the authors claimed that Arl15 binds GTP with the Kd of 20 uM, Arl15 should be mostly GTP-bound in vivo since the cellular concentration of GTP is ~ 0.2 mM, which is 10-fold that of GDP. Therefore, the structure of nucleotide-free Arl15 complexed with the CBS-pair domain probably does not reflect their in vivo binding.

We agree that the nucleotide-binding status of Arl15 is important and that in cells, it likely has GTP bound. However, that does not change the usefulness of the structure of the ARL15-CNNM2 complex. The presence or absence of GTP does not change the binding affinity. This confirms that the conformation of the binding site residues is independent of GTP binding. Most importantly, the structure allowed the identification of a mutation in ARL15 that blocks binding. This mutation disrupts ARL15 inhibition of CNNM and TRPM7 function in cells.

3) Figure 2. The schematic domain organization diagram of CNNMs is too simplified. There are three transmembrane domains. Most importantly and highly relevant to this study, the CBS-pair domain consists of CBS1 + CBS2. Figure 2E should present multiple rotational views of the Arl15-CBS complex with clear labeling of CBS1, CBS2, and Arl15's switch I and II regions. In Figure 2, it would benefit the readers if the authors also present primary sequences of the CBS-pair domain and Arl15 with key amino acids annotated.

We apologize for the confusing presentation. To accommodate more detailed representations of the structure, we have separated the ITC measurements and the crystal structure into two figures. We now show the primary sequences for ARL15 and CNNM2 along with residues mutated in Figure 3.

Balancing clarity and completeness, we have chosen to keep the schematic of the CNNM2 domain simple, only representing complete domains. CNNMs does contains three transmembrane helices but these only form a single transmembrane domain. Similarly, the reviewer is correct in noting that the CBS-pair domain consists of two CBS motifs; however, these fold together and are never found separately (see the review by Ereno-Orbea et al., Arch Biochem Biophys, 2013). In the 3D structures, the CBS motifs intertwine (see Author response image 2). It's not practical to separately label the motifs in the 3D structures but we have highlighted the CBS1 and CBS2 motifs in the primary sequences in Figure 3B.

**Author response image 2. sa2fig2:** CNNM2 CBS-pair domain color-coded, blue-to-red from the N- to C-terminus. The CBS1 (blue to green) and CBS2 motifs (yellow-green to red) fold together to form a single protein domain.

As requested, we now show the complex in two orientations (Figure 3C) and have included a supplemental video to show the arrangement of the domains. We've also added a comparison of the primary amino acid sequences of ARL15 and related GTPase as a supplement to Figure 3.

4) The authors might comment on the following questions in the Discussion. How does Arl15 regulate the activity of TRPM7? Is it via a direct interaction between Arl15 and TRPM7? Or is it via CNNMs? How do CNNMs transduce the signal from Arl15 to TRPM7?

It is tempting to speculate but we can't make any firm conclusions about the mechanism of ARL15 inhibition, other than a requirement for the CNNM-binding site on ARL15. We have updated the Discussion to include ARL15 inhibition of CNNM Mg^2+^ efflux.